



# Insights on estimating urban $CO_2$ emissions using eddy-covariance flux measurements

Kyung-Eun Min[1*], Junphil Mun[1], Begie Perdigones[1], Soojin Lee[1], Kyung-Hwan Kwak[2]

[1]School of Earth Sciences and Environmental Engineering, Gwangju Institute of Science and Technology (GIST), Gwangju, 61005, Korea
[2]School of Natural Resources and Environmental Science, Kangwon National University, Chuncheon, 24341, Korea

*Correspondence to*: Kyung-Eun Min (kemin@gist.ac.kr)

**Abstract.** Living in an era of government allocated carbon dioxide ($CO_2$) emissions, knowing the accurate amount of human-induced $CO_2$ becomes very critical. To this end, an in-depth understanding of $CO_2$ emissions in urban areas where human activities are concentrated will be of practical help. With this motivation, we quantify $CO_2$ emission strengths of individual urban activities (i.e. vehicle, industry, heat generation, etc.) based on direct observations of vertical $CO_2$ exchanges at urban-atmosphere interface using Eddy-Covariance (EC) method at Gwangju, Korea (2017.11-2018.08). Day of week difference analysis, together with varying wind sector, grounded from carefully designed measurement set-up, enables us to assess $CO_2$ emission factors (*EF*s) free from seasonal bias (i.e. heating and urban vegetation); evaluated *EF*s of traffic from day of week difference was 0.017($\pm$0.011) µmol m$^{-2}$ s$^{-1}$ car$^{-1}$ which is more than 10 times larger than that from simple relation (0.0012$\pm$0.0011 µmol m$^{-2}$ s$^{-1}$ car$^{-1}$) between $CO_2$ flux and traffic counts. The $CO_2$ emissions due to the car manufacturing industry within the fetch and heating when air temperatures were lower than 18 °C were quantified as 103.25($\pm$42.18) µmol m$^{-2}$ s$^{-1}$ and 2.41($\pm$1.71) µmol m$^{-2}$ s$^{-1}$ °C$^{-1}$, respectively. Urban vegetation uptake was estimated as -1.72 kg C m$^{-2}$ yr$^{-1}$ only with *EF*s traffic inferred from day of week difference indicating possible erroneous estimation in simple relation unless it properly reflects representative seasonal changes in a year. Even though our estimations are conservative *EF*s due to limitations in corrections of horizontal seepage and vertical storage, we found that both *EF*s for traffic and heat in latest emission inventory were more than 2.5 times lower than our estimations which indicate the urgency in bottom-up inventory validations.

## 1 Introduction

As a mitigation strategy to climate change, most countries in the world are living under the motto of reduction in greenhouse gas (GHG) emission based on global consensus on seriousness and urgency in climate change. In order to realize this, many great deals of efforts are being invested not only reducing carbon dioxide ($CO_2$) emissions from individual to international levels but also re-recognizing $CO_2$ emissions as an economic concept via trading, taxing and capping, since this molecule comprises up to 65 % of radiative forcing among human-driven emissions (NOAA, 2018). Thus precise and accurate





knowledge on anthropogenic $CO_2$ sources and their emission strengths would be beneficial for establishing effective implementation strategies.

As supportive efforts on this aspect, many different methods have been developed to evaluate $CO_2$ emission rates, especially in urban settings, where strong signatures of human activities in a society are concentrated due to urbanization (Coutts et al., 2007). Roughly, 30 to 40 % of anthropogenic $CO_2$ are emitted in urban areas (Satterthwaite, 2008).

One of the most common methods to compile $CO_2$ emission inventory is bottom–up estimation. This method estimates $CO_2$ emission from various source sectors by adding up each element that emits $CO_2$ with their emission strength weighted by the degree of vitality in activity; in most countries worldwide, $CO_2$ emissions from the major factories, automobiles and heating are indirectly compiled via this method based on raw material consumption, traffic volumes, vehicle miles travelled, and energy consumption (Gately et al., 2013; Kennedy et al., 2009; Velasco et al., 2009). The bottom-up method has a strength

to encompass most of $CO_2$ emission-related activities, thus, local, regional, national, and global scale inventory constructions are possible.

However, this approach often reports low bias in estimation, likely due to missing/unknown source components and/or inaccuracies and incompleteness in emission factors ($EF$s) (Kennedy et al., 2010; Leip et al., 2018) as well as uncertainties in the degree of activity estimations such as anthropogenic activity and land cover (Zhang et al., 2014). For example,

unreported land use and/or activities can alter the strength of emission drastically than what is reflected in the bottom-up inventory (Velasco and Roth, 2010). In addition, fixed $EF$s of vehicle have challenges to reveal true emission during real driving conditions with varying traffic situations as the air–fuel ratio changes as described by Lee et al. (2013). Thus, validations of existing bottom-up inventory to assess its legitimacy via independent methods such as inversion-analysis and/or direct $CO_2$ flux measurements are required (Hirano et al., 2015).

There are other indirect means to estimate $CO_2$ emissions such as usage of household expenditure to examine its relationship with energy use and the associated carbon emissions (Druckman and Jackson, 2009; Isaksen and Narbel, 2017). Nighttime light data has also been proven to have advantages in calculating $CO_2$ emissions at finer spatial resolutions (Doll et al., 2006; Ghosh et al., 2010; Lu and Liu, 2014; Shi et al., 2016). Both are beneficial approaches for estimating gross scale $CO_2$ emission. However, they are still limited for activity-specific emission strength estimation, in addition to the uncertainties in

activity measures such as expenditure surveys (Kerkhof et al., 2009) and nightlight data due to sensor saturation in urban cores (Zhang et al., 2017).

Direct observation-based estimation methods are also used. Berkeley Environmental Air-quality & $CO_2$ Network (BEACO2N) is a relatively new approach using dense networks of small observing nodes to measure urban $CO_2$ mixing ratio and other atmospheric gases (Shusterman et al., 2018; Turner et al., 2016). Recently, Turner et al. (2020) estimates urban

$CO_2$ emission from observation changes before and during the COVID-19 (Coronavirus Disease 2019) mobility regulations with an aid of atmospheric transport model, since the observing nodes only provide information of mixing ratio rather than the flux of $CO_2$.





Eddy Covariance (EC) flux measurement is a promising method to validate carbon emission inventories, since it directly provides observation-based $CO_2$ exchanges in urban canopy scale. The $CO_2$ flux measurements based on EC technique have been utilized to elucidate the degree of natural and anthropogenic carbon transfer across the atmosphere and adjacent surface system (Baldocchi et al., 2001; Sabbatini et al., 2018). EC uses covariance of vertical wind speed and targeted scalar concentration to evaluate the degree of exchanges based on high time resolution observations in precision to capture all signatures in full suite of eddy scales which ranges from sub-seconds to hours (Desjardins, 1974; Katul et al., 2001; Liang and Wang, 2020). This technique was originally developed to evaluate the plant productivity over homogeneous plano-surface (Baldocchi et al., 2001 and references therein), however, its application has been expanded from traditional to the non–ideal environment i.e. urban area (Christen et al., 2011; Coutts et al., 2007; Crawford et al., 2011; Grimmond et al., 2002; Lietzke et al., 2015; Moriwaki and Kanda, 2004; Nemitz et al., 2002), and the number of measurements has grown rapidly over the past two decades (Menzer and McFadden, 2017; Björkegren and Grimmond, 2018; Conte et al., 2018; Goret et al., 2019; Stagakis et al., 2019; Rana et al., 2021; Matthews and Schume, 2022).

Applications of EC technique in an urban setting have a strength in $CO_2$ flux monitoring, since it is capable to capture the signature of exchanges in hours to years in time span (Wofsy et al., 1993) with spatial range which varies from few hundred meters to kilometers (Schmid, 1994) and thus easily covers a city scale. For this reason, several attempts have been made to estimate urban $CO_2$ emission, as summarized in Table 1. The reported urban $CO_2$ flux in Table 1 largely varies, but it is possibly due to various environmental settings (e.g. population, traffic, vegetation density as well as its maturity, etc.) where the measurements had been conducted. Generally, urban sites are characterized as net source of $CO_2$, reported with positive sign in flux.

**Table 1.** Summary of reported urban/suburban $CO_2$ flux measurements by eddy covariance technique, together with site characteristics such as land use type and mean canopy and measuring heights ($z_h$ and $z_m$). Emission factors (*EF*s) for heating and traffic are also listed, if available.

| City, Country | Land Use Type | $z_m(z_m/z_h)$ [m] | Flux [$\mu$mol m$^{-2}$ s$^{-1}$] | EFs Heating [$\mu$mol m$^{-2}$ s$^{-1}$ °C$^{-1}$] | Car [$\mu$mol m$^{-2}$ s$^{-1}$ car$^{-1}$] | Reference |
|---|---|---|---|---|---|---|
| Arnhem, Netherlands | Urban | 23 (2.1) | 5.76 | 1.483[*] | 0.075[*] | Kleingeld et al., 2018 |
| Basel, Switzerland | Urban | 41 (2.5) | 16.4 | 0.019[*] | 0.011[*] | Lietzke et al., 2015 |
| Beijing, China | Urban | 47 (2.8) | 13.0 | 0.34[*] | | Liu et al., 2012 |
| Cairo, Egypt | Urban | 35 (1.6) | 6.18 | | | Burri et al., 2009 |
| Chicago, USA | Suburban | 27 (4.29) | 3.67 | | | Grimmond et al., 2002 |
| Essen, Germany | Urban | 26 (1.7) | 9.30 | | | Kordowski and Kuttler, 2010 |





| | | | | | |
|---|---|---|---|---|---|
| Firenze, Italy | Urban | 33 (1.32) | 25.8 | | Matese et al., 2009 |
| Helsinki, Finland | Urban | 26 (3.7) | 4.78 | 0.004 | Vesala et al., 2008 |
| Heraklion, Greece | Urban | 27 (2.4) | 6.77 | 9.66 $\mu mol\ m^{-2}\ s^{-1®}$ | Stagakis et al., 2019 |
| Houston, USA | Urban | 60 (10.5) | 8.00 | 0.002* | Park and Schade, 2016 |
| Lecce, Italy | Urban | 14 | 7.73 | | Conte et al., 2018 |
| Łódź, Poland | Urban | 37 (3.4) | -5 to 15 | 0.183-3.499* | Pawlak et al., 2011 |
| London, UK | Urban | 190 (22) | 38.0 | a: 4.05-6.89 b: 4.09-5.47×10⁻⁴** | Helfter et al., 2011 |
| London, UK | Urban | 49 (2.2) | 33.6 | 1.95* | 6.7, 5.2 $\mu mol\ m^{-2}\ s^{-1†}$ | Ward et al., 2015 |
| London, UK | Urban | 46.4 (2.21) | 37.04 – 38.56 | 27.39 $\mu mol\ m^{-2}\ s^{-1®}$ | Björkegren and Grimmond, 2018 |
| Melbourne, Australia | Suburban | 40 (3.33) | 6.12 | | Coutts et al., 2007 |
| Mexico City, Mexico | Suburban | 37 (3.8) | 19.5 | 1.53-3.87* | Velasco et al., 2005 |
| Montreal, Canada | Urban | 25 (3.16) | 10.17 | | Bergeron and Strachan, 2011 |
| Münster, Germany | Urban | 65 (4.33) | 12 | | Schmidt et al., 2008 |
| Osaka, Japan | Urban | 127 (11.9) | 1.29 -12.9 | 0.26* | Ueyama and Ando, 2016 |
| Tokyo, Japan | Suburban | 29 (4) | 5.68 Summer 13.0 winter | | Moriwaki and Kanda, 2004 |
| Vancouver, Canada | Urban | 24.8 (3.82) | 17.0 | 1.45* | 11.71 $\mu mol\ m^{-2}\ s^{-1¶¶}$ | Crawford and Christen, 2015 |
| Vienna, Austria | Urban | 144 (5.8 – 9.6) | 13.97 | 0.72* | 0.0089, 0.0043© | Matthews and Schume, 2022 |
| **Gwangju, Korea** | **Urban** | **88 (7.1)** | **28.26** | **2.45** | **0.017§** | **This study** |

\*: simple linear regression; ®: rely on road fraction from land cover than number of cars; \*\*: exponential regression expressed as Flux = $ae^{b\text{Cars}}$; †: traffic counts are not considered but weekdays and weekend fluxes, ¶: unit conversion from daily emission; ©: winter and summer weekday; §: traffic counts difference in weekdays and weekend



Relatively, less attempts are made to infer $CO_2$ $EF$s from EC flux measurements compared to its spatiotemporal analysis which limit the usage of EC flux measurements. Table 1 also lists few studies which evaluated $EF$s for heating ($EFs_{heating}$) and traffic ($EFs_{traffic}$) from their $CO_2$ EC flux measurements. $EFs_{heating}$ were inferred from the relationship between observed $CO_2$ fluxes and temperatures over the heating degree days (HDD, Crawford and Christen, 2015; Kleingeld et al., 2018; Lietzke et al., 2015), while $EFs_{traffic}$ were mostly evaluated from simple linear relation (here after SLR; Järvi et al.,

2012; Kleingeld et al., 2018; Lietzke et al., 2015; Matthews and Schume, 2022; Nemitz et al., 2002; Park and Schade, 2016; Velasco et al., 2009) or power-correlation (Helfter et al., 2011) between traffic volumes and $CO_2$ fluxes.

Though the yearly averaged $EFs_{traffic}$ likely reduce complexities in modelling, this method is susceptible to the seasonal bias owing to variations in natural and anthropogenic $CO_2$ emissions such as changes in photosynthesis and respiration by urban vegetation, space heating, degree of incomplete combustion of vehicles, etc. as Helfter et al. (2011) described with

varying $EFs_{traffic}$ among seasons. Thus, more than a year of measurement is required to properly integrate the seasonal changes in $EF$s with careful consideration in analysis for skewness in the number of observations in each season. In some regions of the world, measured number of data imbalance among seasons are unavoidable due to the inherent natural variabilities especially vulnerable with open path sensors during rainy seasons.

As an alternative to this aspect, we suggest an analytical approach to extract $CO_2$ $EF$s among the differences in day of week

(DOW) pairs from EC flux observations not only to minimize the seasonal bias but also to enable $EF$s evaluations with relatively short-term period than many years of measurements. From this practice, we were able to see the subtlety in inferring the magnitude of urban vegetation uptakes as well as the importance in the seasonal bias free $CO_2$ $EF$s, in addition to the comparison with existing emission inventory.

## 2 Methods

### 2.1 Site description


To directly assess the $CO_2$ $EF$s from various sources in an urban area, EC system was set up and operated at Gwangju in Korea (35° 10' 0.0048" N, 126° 54' 59.9904" E) from November 2017 to August 2018 (Figure 1) just before the start of the typhoon season. This city is a legitimate test bed for anthropogenic $CO_2$ $EF$s estimation, not only due to its high population density (2,920 people km$^{-2}$) with various land use types (commercial, residential and recreational area with industrial

facilities as well, information available from the website of the Environmental Geographic Information Service, EGIS, https://egis.me.go.kr/main.do) but also due to its flat terrain especially in the central area where the City Hall is located. In addition, this city setting is even more ideal for the aim of this study since the Gwangju Industrial Complex, which hosts an automobile production plant, is located 2 km away from the city hall in eastern direction (Figure 2a).



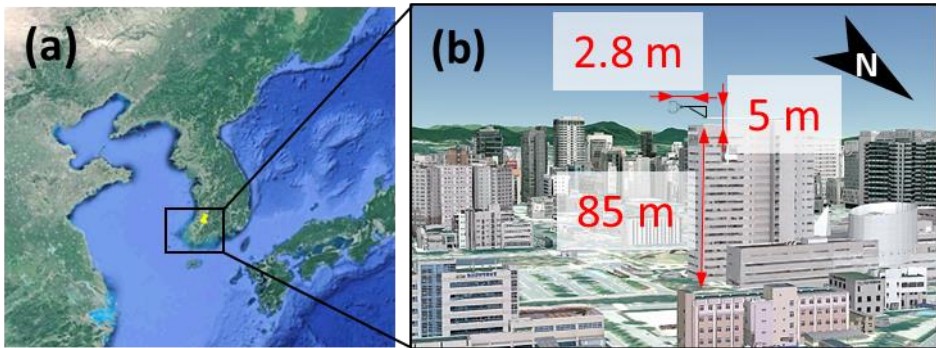

**Figure 1.** (a) Location of the study area in Gwangju, Korea, © Google Maps 2022 and (b) instrumental set-up position on top of the Gwangju city hall building visualized in 3-dimensional view provided by the geospatial information open platform, Vworld.

The $CO_2$ flux measuring system was installed on the helideck of the Gwangju city hall (90 m above the ground – building height: 85 m, helideck: 3 m and measuring system structure: 2 m, shown in Figure 1b) to fulfill the height requirements for EC flux measurements. Roth et al. (2017) suggested that the flux measuring height should be at least 2 to 4 times higher than the canopy height. The mean canopy height($z_h$, 12.33 m) of the study area is determined from the building height data available from the Electronic Architectural administration Information System (EAIS, https://cloud.eais.go.kr/). The required information for the footprint analysis were extracted as roughness length ($z_0$) of 1.6 m and displacement height ($z_d$) of 9.1 m from the inferred relation of roughness elements density and $z_0/z_h$ and $z_d/z_h$ by Grimmond and Oke (1999).

Our EC system was installed outside of inertial sublayer (average of 25 m within footprint area and 80 m near the city hall, following Grimmond and Oke, 1999) and sufficiently lower than the planetary boundary layer (Bottema, 1997) which varies from 0.3 km to 1.2 km at this site, determined from the balloon-borne measurements of potential temperature profiles acquired by Republic of Korea Air Force at Gwangju airport (4 times in a day, 3:00, 9:00, 15:00, and 21:00, local time, from 2001 to 2018).

To minimize the wind distortion caused by the city hall building, the EC flux measuring system was installed on the helideck and outreached by 2.8 m outside of the building in a diagonal direction as shown in Figure 1b. Further interpretations of flux measurements are limited to the direction where the wind distortions were minimized, based on Computational Fluid Dynamics (CFD) calculation (Figure S1).

**2.2 Measurement method**

We used a 3-D sonic anemometer (CSAT3B, Campbell Scientific Inc., Utah, USA) and an open path $CO_2$ and $H_2O$ analyser (LI-7500RS, LI-COR, Nebraska, USA) operated at 10 Hz for the flux measurements. From the data sets, half hour fluxes of $CO_2$ and $H_2O$, as well as latent and sensible heat, were calculated based on EC method as shown in equation (1), using the SmartFlux 2 (LI-COR) software,

$$F = \overline{w'c_x'} \tag{1}$$





where $F$ represents the vertical flux of individual scalar ($x$) which can be evaluated from the averages (shown as an upper bar in equation 1) of the covariance (represented as $'$ in equation 1) of vertical wind velocity ($w$) and the abundance of $x$ ($c_x$), under the assumptions that mean vertical wind speed is zero and the changes in air density is negligible.

As the first pre-processing step, raw data were de-spiked; acceptable ranges of $c$ and $w$ were defined as ±3σ and ±5σ from the mean, following Vickers and Mahrt (1997). The outranged observed $c_{co2}$ and $w$ were treated as spikes, thus replaced with mean for further analysis. Then, 2-D axis rotation was performed to make the average vertical wind speed become zero ($\overline{w} = 0$) with double rotation method as Kaimal and Finnigan (1994). Calculated fluxes were averaged every 30 min with the use of 5 min block average to reflect the slow changes in $CO_2$ concentration. Then, Webb–Pearman–Leuning (WPL) correction (Webb et al., 1980) was applied to compensate for the effect of air density fluctuation.

As a quality control (QC) scheme, steady state and turbulence characteristic tests were performed by following Mauder and Foken (2004) to filter out the periods of drastic $CO_2$ concentration changes and too stagnant condition where EC assumption breaks. Only the 30 min flux data points which fall in the interquartile range of 6 continuous 5 min fluxes were used, and this results in 18.3% data reduction. Furthermore, friction velocity threshold ($u_*$>0.2) was applied only for accounting sufficient turbulent condition, which results in 9.4% data rejection. Due to the intrinsic nature of open-path nondispersive infrared (NDIR) sensor for high frequency $CO_2$ measurement, data influenced by precipitation and high relative humidity (>90%) were removed (3.1%). Also, data points with sharp and/or large changes in rotation angles (more than 3° of rotation) were filtered out (7.8%) to exclude measurements under agitating condition owing to external physical factors. With all the filtering processes, 37% of data has been discarded.

**2.3 Footprint analysis**

To apportion the $CO_2$ sources in flux measurement, footprint analysis was conducted using two different approaches- Kormann and Meixner (2001) and Kljun et al., (2004) models to process non-neutral condition and to evaluate broader ranges of boundary layer stratifications. Here, footprint boundaries were defined to confine 70% of average total flux during the measurement period. Detailed calculations and parameterizations are described in S2 of the Supporting Information. Fetches were drawn with 5° bins, starting from north as 0° with increase in clockwise direction, centred at the city hall as shown in Figure 2a. The atmospheric stability was also evaluated from the length scale ($\zeta$) with Obukhov length ($L$) calculation; unstable (stable) condition is determined as when $\zeta = (z_m - z_d)/L$ is lower (higher) than -0.05 (0.2). Other conditions are classified as near neutral condition. Site characteristics regarding atmospheric stability are described in S2.

To assess the quantitative contributions of the individual sources, the wind directions were split into two sectors; (1) the Eastern Industrial Area (EIA, 45°-100°) and (2) the Southern Green Area (SGA, 100°-225°), based on whether or not the fetch includes the automobile production plant and urban vegetation (Figure 2).





### 2.4 Emission factor estimation

**2.4.1 Vehicular CO₂ emission**

Flux of CO₂ ($F_{CO2}$) measurements over SGA were used to estimate $EF_{traffic}$ for minimizing the influence of industrial CO₂ emission by comparing inferred traffic counts and $F_{CO2}$. SLR and DOW difference methods were used for $EF_{traffic}$ to gauge the influence of seasonal changes in anthropogenic and biogenic CO₂ emissions. The distinct difference in DOW $F_{CO2}$ provides adequate information for $EF_{traffic}$ with minimum seasonal biases, since the photosynthetic activities and space heating hardly change between weekend and weekdays within a week frame. Hence, the ratio of the flux and traffic count difference ($\Delta F$ and $\Delta T$) between weekday and weekend were used as $EF_{traffic}$ (Eq. 2).

$$EF_{traffic\_DOW} = \frac{\Delta F}{\Delta T} = \frac{F_{weekday} - F_{weekend}}{T_{weekday} - T_{weekend}} \qquad (2)$$

Since no in situ traffic counts were available near the city hall, it was inferred from the measurements of inbound and outbound of 6 highway tolls surrounding Gwangju (Gwangju, Donggwangju, Hakun, Songam, Seogwangsan and Donggwangsan), together with the occasional observations on the adjacent crossroads on the eastern and southern side of the city hall (Gyesu and Sangmu district), in hourly resolution. The locations of tolls and crossroads are shown in Figure S3. The highway toll data are available from Korea Expressway Corporation's public data portal (http://data.ex.co.kr/) and occasional survey information are available upon request to Gwangju Metropolitan City Transportation Department (https://www.gwangju.go.kr/traffic/).

For 30 minute traffic counts, under the assumption of even distribution of traffic within an hour, averages of ratios between adjacent crossroads and highway tolls were used not only over the study period but also for the years of 2019 and 2020 (2017-2020 and 2019-2020 for weekdays and weekend) since weekend and Friday data were only available after 2019. Since weekday (weekend) traffic survey only happened from Tuesday to Thursday (Sunday), Monday and Friday (Saturday) ratios inferred from weekday averages (Sunday only) were not used for $EF_{traffic}$ but $EF_{industry}$. Detailed descriptions on diurnal traffic patterns with respect to days in a week are in supporting information S3. Briefly, distinct bimodal patterns which peaked in the morning and afternoon rush hours were observed (as also seen in Figure 5a) for the weekdays with slight difference in Monday and Friday (Figure S4). Meanwhile, weekend patterns showed delays in morning peaks with broader afternoon peaks (Figure 5a) and Sunday was used as weekend. To maximize the number of samples for DOW difference, two pairs of weekend-weekdays were made, one in the previous week and one after that which centered around a specific Sunday.

Time window of 7:00 to 10:00 was used as a default morning hours and ranges from 6:00 to 10:00 with varying averaging window size of 1 to 4 hours were applied as robustness tests in $EF_{traffic}$ estimations. The linear correlations between traffic counts and CO₂ fluxes were extracted considering uncertainties in both scalars defined as 1 standard deviation in binning window, and the errors in parameters were computed using Monte Carlo simulation assuming Gaussian distribution. For





convenience, $EFs_{traffic}$ obtained through the day of week difference are named as $EF_{traffic\_DOW}$, in order to distinguish with that from the SLR method ($EF_{traffic\_SLR}$).

One should note that the inferred traffic counts should be treated as an activity index which represents the relative amount of car fleet than the actual integrated number of cars within the fetch. Thus special treatments are required to compare the $EFs_{traffic}$ with other studies, since the degree of closeness in traffic volume with respect to the total traffic counts in a footprint likely varies among studies.

### 2.4.2 Industry and heat related CO₂ emission

The data from EIA were used to evaluate the emission factor from industrial area ($EF_{industry}$). The $CO_2$ emissions from on-road vehicles, inferred from the previous section, were subtracted from the EC flux measurements to extract the industrial emission. Even with the difference in contribution of other $CO_2$ emission/uptake sources than traffic and industry in EIA and SGA, DOW $CO_2$ fluxes difference owing to them is expected to be negligible, since their emissions/uptakes hardly change within a week. The traffic density difference between SGA and EIA was corrected based on the ratio of the traffic survey of two representative crossroads.

In reality, the $CO_2$ emission from the car manufacturing company only happens over the factory site (0.298 km$^2$), therefore, extracted $EF_{industry}$ was calculated by normalizing the areal coverage of the facility against the whole area in EIA (4.32 km$^2$).

As a conservative estimation in $EF_{industry}$, the time window of 11:00 to 14:00 was chosen not only because the normal operation hours are 6:00 to 22:00 during weekdays but also because the traffic volume changes with respect to time are relatively low, and this minimizes error accumulation. As a sensitivity test, different time windows from 10:00 to 14:00 with varying averaging ranges of 2 to 4 hours were used.

To gauge the amount of $CO_2$ emission related to space heating during the low temperature season, analysis via HDD estimation as by Kleingeld et al. (2018) was performed. Briefly, HDD was calculated by multiplying the heating temperature and the number of days when the temperature was below 18 ℃, assuming that the heating tendency stops once the temperature reached 18 ℃. For September and October in 2018, inferred HDD with second order regression were used (purple line in Figure S5). Both EIA and SGA data were considered for $EFs_{heating}$ with time window of 10:00 to 14:00, when the $CO_2$ fluxes showed clear difference above and below 18 ℃ with light traffic condition. The $EF_{heating}$ was estimated from the slope of $CO_2$ fluxes and temperatures under the temperature limit and the sensitivity tests were conducted by varying the threshold from 10 to 22 ℃ with 2 ℃ bins.

### 2.4.3 Year round CO₂ emission and vegetation contribution

In this section, we describe how we scale up to evaluate the yearly emission of $CO_2$ among activities in city scale to compare it with Gwangju greenhouse gas emission inventory of 2018 (available from International Climate & Environment Center,





2018). This inventory provides the emissions of $CO_2$ by the lowest administrative district of urban, named as "Dong" in Korea; a total of 10 dongs were overlapped within the footprint. Since the district boarders mismatched with the footprint boundary, the normalized emissions by area were compared.

To calculate the annual $CO_2$ emission in city scale, areal coverage of individual activities was considered. Since the on-road vehicle emission happens all over the fetch due to the densely populated traffic pathways, half hour varying median traffic volume (4,023 cars) inferred from crossroad surveys was used with areal coverage of road (impervious, 5.32 km²). For the total $CO_2$ emission from heat, $EFs_{heating}$ were considered within total area of residential, commercial and other buildings in the fetch (3.10 km²) with HDD of 2,330 °C in a year.

## 3 Results and discussion

### 3.1 Footprint and land use type analysis

Analysed footprints by following Kormann and Meixner 2001 (in blue) and Kljun 2004 (in red) are shown in Figure 2a which stretched to roughly 3.5 km radius in radial direction. Large differences are observed in southwest to northeast region in clockwise direction (shaded with red and blue), which likely originated from wind distortion owing to the structure of the

building and helideck. Meanwhile, the fetches from northeast to southwest side agree well; based on the CFD models, less than 20% of wind speed distortions were observed for the wind blowing from 45° to 225° (shown in Figure S1). Thus, for the rest of the analysis, the averaged footprint between two methods with wind sector within the range of 45°-225° was used.

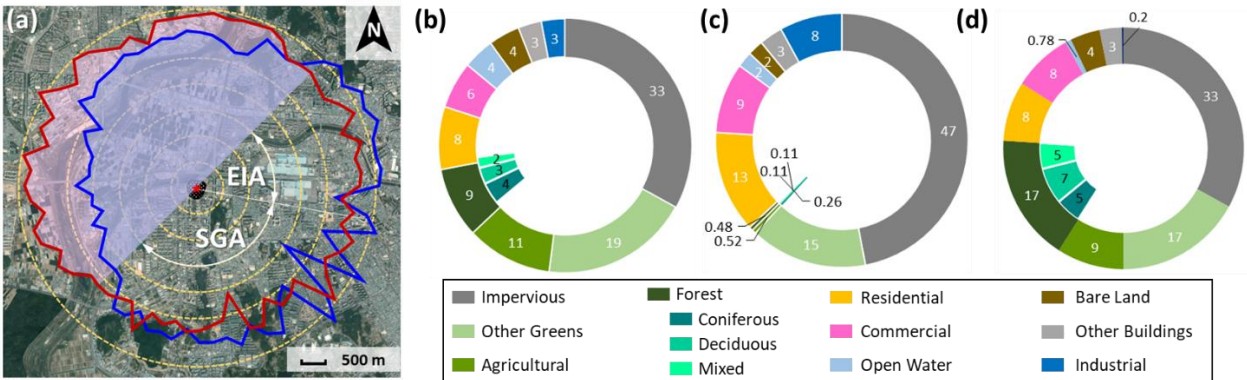

**Figure 2.** (a) Footprint analyses on the land use map (red and blue lines by Kljun (2004) and Korman and Meixner, (2001), respectively)
with the fraction of land use types in percentage for (b) all radial direction, (c) EIA (Eastern Industrial Area) and (d) SGA (Southern Green Area) sector. The red and blue shaded regions in (a) were not used. Black patched area and red stars in the center area of (a) represent the invisible region owing to the sensor height and location of city hall. © Google Maps 2022

The land use type within the fetch is shown in Figure 2. The complex land use character in the central area of Gwangju
within the footprint (Figure 2b) ranges from impervious road and parking lot (34.2%), greens (39% including forests, agricultural areas of rice paddy, farm field and orchards as well as other greens of yards and graves), residential (7.80%),





commercial (6.22%) to industrial (2.64%). Figure 2c and d further categorize land use components for EIA and SGA; the major differences are whether the area includes industrial area or not (EIA: 7.87% vs SGA: 0.21%) as well as the fraction of forest and agricultural area (EIA: 1% vs SGA: 26%) except the fraction of yards and graves marked as other greens.

### 3.2 CO₂ measurement descriptions

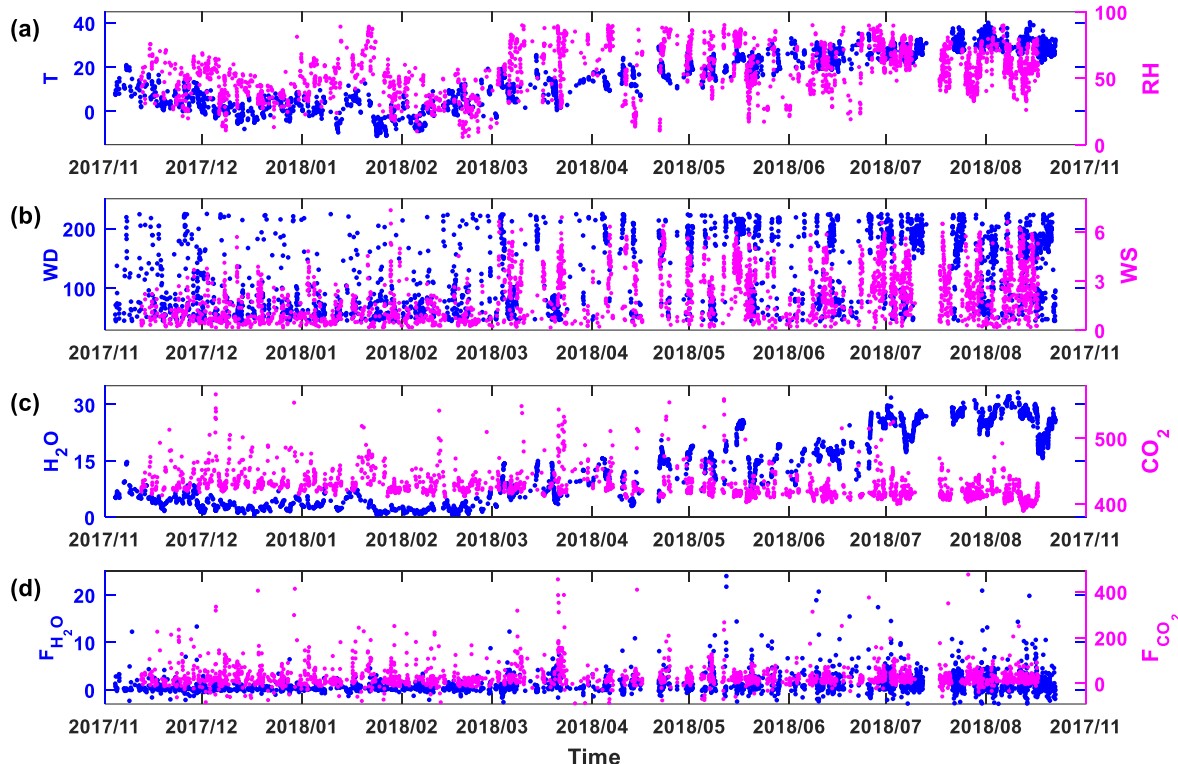

**Figure 3.** Time series of (a) temperature (T, ℃) and relative humidity (RH, %), (b) wind direction (°) and speed (m s⁻¹), (c) mixing ratios of $H_2O$ and $CO_2$ (ppth, part per thousand, and ppm, part per million), and (e) fluxes of $H_2O$ and $CO_2$ ($F_{H2O}$ and $F_{CO2}$, **mmol m⁻² s⁻¹** and **µmol m⁻² s⁻¹**, respectively).


Time series of wind sector filtered temperature (T), relative humidity (RH), wind direction (WD), wind speed (WS), mixing ratios and fluxes of $H_2O$ and $CO_2$, ($H_2O$, $CO_2$, $F_{H2O}$ and $F_{CO2}$, respectively) are shown in Figure 3. The averages (interquartile range) of T, RH and WS were 17.17(±10.57) °C, 56.32(±15.70) % and 2.36(±1.28) m s⁻¹, respectively. The observed mixing ratios of $H_2O$ and $CO_2$ were 14.57(±7.45) ppth (part per thousand) and 425.58(±13.00) ppm (part per million) and their fluxes were 1.12(±0.68) mmol m⁻² s⁻¹ and 28.26(±14.14) µmol m⁻² s⁻¹, respectively.

T, RH and $H_2O$ mixing ratio clearly showed seasonal changes in March and June from winter to spring and spring to summer. After July, monsoon starts as indicated by $H_2O$ abundance and RH. Measured $F_{H2O}$ showed larger variations after March than wintertime, likely due to active evaporation processes as temperature increases. As opposed to $F_{H2O}$, observed





$F_{CO2}$ had more variation in cold than warm season, mainly due to the strong vertical concentration gradients reflecting the
active changes in $CO_2$ emission/uptake processes (i.e. space heating, respirations, etc.). The $F_{CO2}$ with positive sign indicates
that Gwangju is a net $CO_2$ source like other cities in the world as listed in Table 1, and comparable to Firenze and London
(Matese et al., 2009; Ward et al., 2015). The observed $F_{CO2}$ range agrees well with previously reported $CO_2$ emission
estimation (yearly average of $26.6\pm11.2$ µmol m$^{-2}$ s$^{-1}$) by expenditure pattern (Moran et al., 2018).

However, our direct measurements of $F_{CO2}$ showed clear seasonal variabilities (Figure 3 and Figure S6). Monthly changes in
$F_{CO2}$ showed distinct patterns in cold (November–February: $26.59\pm18.13$ µmol m$^{-2}$ s$^{-1}$) and warm (April–August:
$17.32\pm11.26$ µmol m$^{-2}$ s$^{-1}$) seasons. Meanwhile, mixing ratio of $CO_2$ gradually decreased from January ($441.28\pm12.85$ ppm)
to August ($421.03\pm10.55$ ppm) and recovered in winter period (November to December: $443.97\pm14.88$ ppm) even though it
is hard to determine when it exactly happened due to the missing data in September and October 2018.

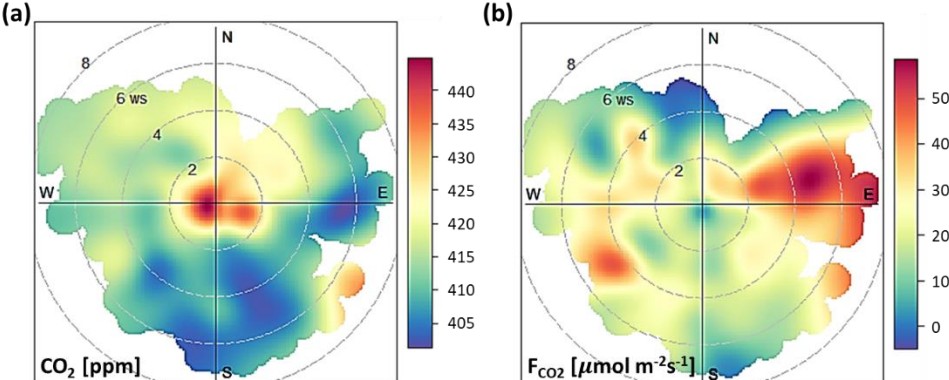

**Figure 4.** Polar plots of (a) $CO_2$ mixing ratio and (b) $F_{CO2}$ centered at where the measurements were conducted.

The polar plots of $CO_2$ mixing ratio and flux are shown in Figure 4, where the color represents wind speed weighted $CO_2$
mixing ratio and $F_{CO2}$. Enhanced $CO_2$ mixing ratio was observed under low wind speed condition indicating strong influence
of main city roads near the city hall where intensive activities take place. Meanwhile, the polar plot of $F_{CO2}$ contrasted with
that of concentration. Strong $F_{CO2}$ signals were observed when wind blew from East-northeast (ENE) to southeast mainly
where the car manufacturing factory exists. On the other hand, the relatively low $F_{CO2}$ signature was observed from South–
southeast (SSE) to south area where the forests are situated. The averaged highest $F_{CO2}$ was observed with wind speed
around 4.7 m s$^{-1}$ from ENE direction in EIA. We would like to clarify that the $F_{CO2}$ data from Southwest (SW) to Northeast
(NW) were not considered to be interpreted due to possible wind distortion by the city hall building.





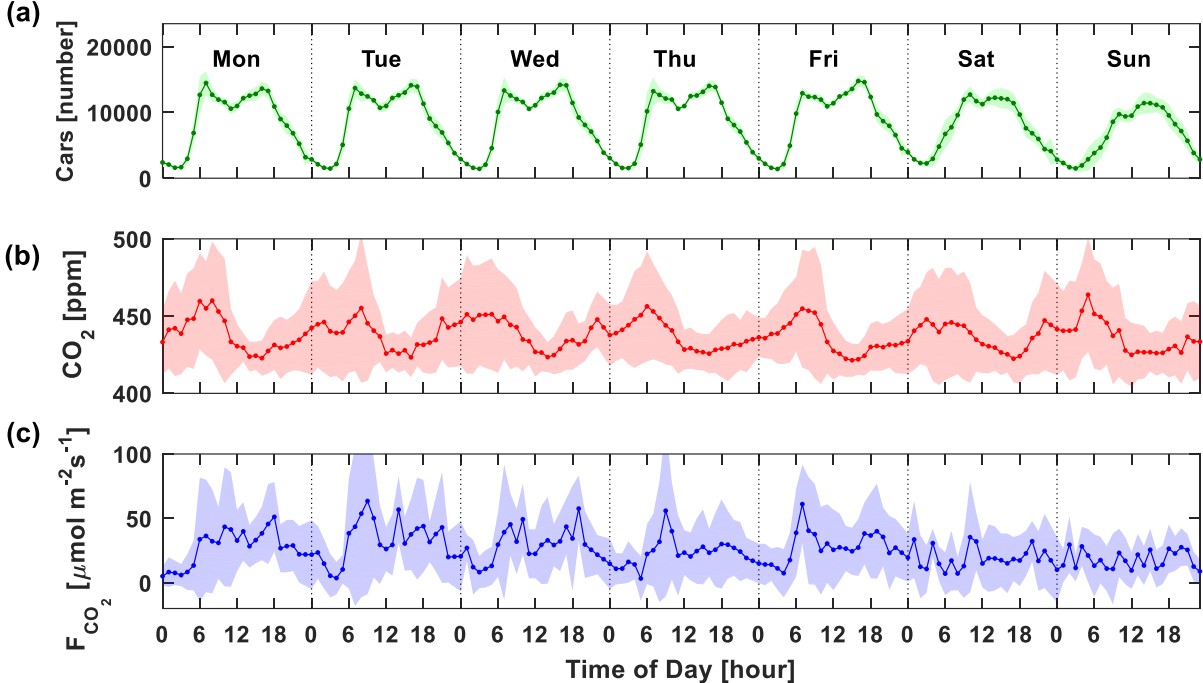

**Figure 5.** Weekly diurnal patterns of (a) traffic volumes, (b) $CO_2$ mixing ratio and (c) $F_{CO2}$. Dotted marker with lines and area represent mean and interquartile range of each factor.

The weekly diurnal patterns of $CO_2$ and $F_{CO2}$ along with traffic counts are shown in Figure 5. $CO_2$ mixing ratio (Figure 5b)

increased until 6:00-7:00 and then decreased until around 18:00 and recovered the next morning, regardless of day of the

week, mainly due to changes in vertical mixing as PBL develops after the sun rises and variations in urban vegetation

activity on top of daily anthropogenic emissions as explained by (Sargent et al., 2018). On the other hand, apparent $F_{CO2}$

difference between weekday and weekend was observed (Figure 5c); weekday flux was higher than that of the weekend. It is

mainly attributed to changes in human activities over a week such as traffic volume related with commuting fleet (shown in

Figure 5a) and operations of the car manufacturing facilities. Apparent consistency in diurnal variations of $F_{CO2}$ and traffic

volume during weekday (Monday to Friday) was observed with two local maxima around 6:00-9:00 and 17:00-19:00 (Figure

S7). Meanwhile, broader enhancement traffic counts with relatively weak or no apparent peaks in $F_{CO2}$ patterns existed over

Saturday and Sunday (also shown in Figure S7). From this relation, $EF_{traffic}$ was extracted as follows.

**3.4 Emission factors ($EFs$) estimations**

With varying wind sectors, observed $F_{CO2}$ can be further analysed to apportion $EF$s of each of the sources under the

assumption of well mixing up to the measurement height. One should note that even with the careful design of instrumental

setup (i.e. located at constant flux layer on flat fetch) and filtration scheme for only considering active turbulent condition,





we cannot rule out the potential $CO_2$ loss by advection as well as accumulation near the surface which could not be detected by our sensor. With limited information in $CO_2$ gradient measurements, horizontal seepage and vertical storage corrections were not performed. Thus, we only provide $EFs$ of lower limit in emission than actual strengths.

### 3.4.1 $EF_{traffic}$ emission

$EF_{traffic}$, from the observations with windblown from SGA using SLR, was quantified as 0.0012(±0.0011, 92% error) μmol m$^{-2}$ s$^{-1}$ car$^{-1}$ (RMSE=3.01) as shown in Figure 6a, which falls in the range of other previous findings; this value is lower than Arnhem (0.075, Kleingeld et al., 2018), Basel (0.011, Lietzke et al., 2015), and Mexico City (1.53-3.87, Velasco et al., 2005) but in similar range with Houston (0.002, Park and Schade,2016), Helsinki (0.006, Järvi et al., 2012), and Edinburgh (0.0017, Nemitz et al., 2002). However, as mentioned in Sect. 2.5.1, $EF_{traffic\_SLR}$ is inherently vulnerable to seasonal variations especially for the case when the measurement cannot represent whole year round seasonality. The intercept of fitted line (shown in Figure 6a) indicating $CO_2$ emission under 0 traffic condition was 4.79 μmol m$^{-2}$ s$^{-1}$ with large variability range of ±12.53 μmol m$^{-2}$ s$^{-1}$ likely referring the various degree of $CO_2$ emission/uptake among seasons and the large uncertainty in regression slope (±0.0011 μmol m$^{-2}$ s$^{-1}$ car$^{-1}$) has likely arisen due to this reason as well.

Meanwhile, $EF_{traffic\_DOW}$ was extracted as 0.017 (±0.011, 65% error) μmol m$^{-2}$ s$^{-1}$ car$^{-1}$ (RMSE=3.66) which is more than an order of magnitude higher than $EF_{traffic\_SLR}$ (Figure 6b) but falls in those of Basel and Mexico City conditions. Even though our number agrees with other works, however, as mentioned in section 2.4.1, simple comparison with $EF_{traffic}$ previous works is not recommended, since the inferred traffic counts only describe the degree of busyness of on-road vehicles in the fetch rather than the actual number of total cars; crossroad surveys are inherently lower than the total amount of traffic in fetch area. Thus the differences in $EFs_{traffic}$ among studies likely arise from the variations in degree of consistency in their traffic volume with total number of cars as well as actual difference in $EF_{traffic}$.

One should note that the intercept and its range of fitted line with DOW method drastically reduced to 2.09 ± 4.34 μmol m$^{-2}$ s$^{-1}$, which is a sign of effective removal in seasonality. However, the cause of non-zero intercept of $F_{CO2}$ indirectly indicates the existence of $CO_2$ emission with weekend and weekday changes in SGA other than traffic. Unaccounted $CO_2$ emissions from industrial, commercial and other activities may explain it, and thus an interesting topic for future investigations. We do not believe that human respiration influences on it due to the implementation of DOW difference in $F_{CO2}$. The sensitivity tests of $EF_{traffic\_SLR}$ and $EF_{traffic_{DOW}}$ showed 13.9 and 27.06% (1 standard deviations) changes, respectively. The uncertainties in $EF_{traffic\_SLR}$ and $EF_{traffic\_DOW}$ were estimated as 92% and 65%, correspondingly from their linear regression fitting uncertainties.




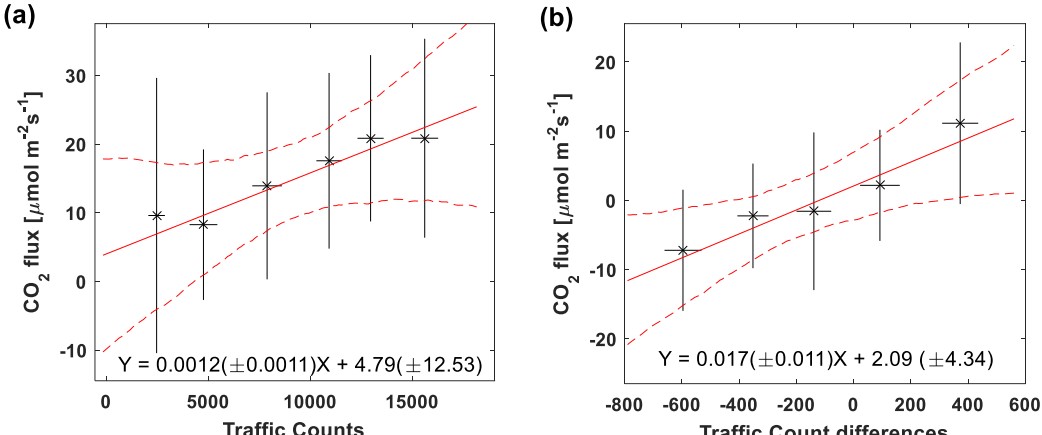

**Figure 6.** Estimation of traffic emission factor ($EF_{traffic}$) from the relation of $F_{CO2}$ with respect to the number of cars, estimated by (a) simple linear regression (SLR) and (b) day of week difference (DOW).

### 350  3.4.2 $EF_{industry}$ and $EF_{heat}$ estimation

Data from EIA were used for $EF_{industry}$ estimation where the measured $F_{CO2}$ contains $CO_2$ emission features of residential, commercial, industrial, traffic as well as urban vegetation. As similar to $EF_{traffic\_DOW}$, by taking the day of week difference in $F_{CO2}$, we intended to ease out the contributions from residential and commercial area as well as vegetation under the assumption that their $CO_2$ emission/uptake patterns are held relatively constant in a week. Figure S8 shows the diurnal

pattern differences in DOW for traffic and $F_{CO2}$ in EIA. From the $EF_{traffic\_DOW}$ and traffic counts in eastern direction, the lower limit of $EF_{industry}$ was estimated as 103.25 (±42.18, 41% error) μmol m$^{-2}$ s$^{-1}$ normalized to the areal coverage (0.341 km$^2$) of car manufacturing company and varied by 54.25 % (1 standard deviation) as the result of sensitivity tests. Considering normal operation hours of 6:00-22:00, from Monday to Friday, yearly emission was assessed as 66 (±34) kt $CO_2$ km$^{-2}$ under the assumption of constant emission over the operation hours except during holidays.





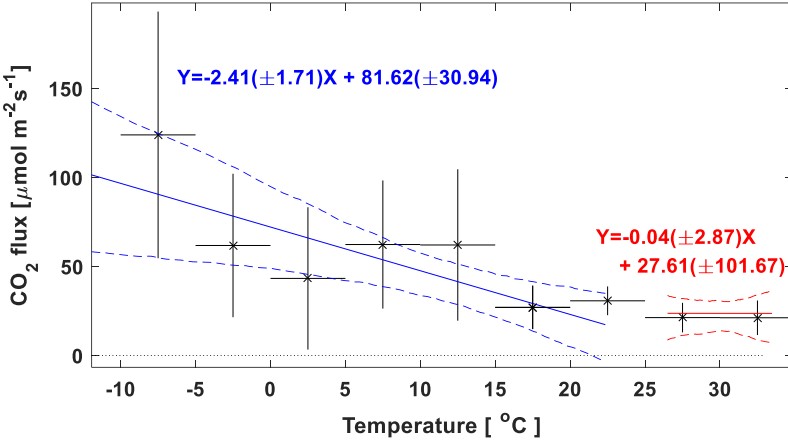


**Figure 7.** $F_{CO2}$ changes with respect to the ambient temperature. Blue solid and dashed lines represent linear regression and confidence band, respectively, for cold period and red lines for warm period.

In regard to heat related $CO_2$ emission, calculated HDD of each month (shown in Figure S5) was used. From the relation
between temperature and $CO_2$ flux shown in Figure 7, where $CO_2$ fluxes tended to decrease as the temperature increased up to 18 °C, the $EF_{heating}$ was estimated as 2.41($\pm$1.71, 71% error) μmol m$^{-2}$ s$^{-1}$ °C$^{-1}$ similar magnitude with those in Arnhem, Łódź and London (Kleingeled et al., 2018; Pawlak et al., 2011 and Ward et al., 2015) and thus year round emission of 21($\pm$15.1) kt $CO_2$ km$^{-2}$ was evaluated.

We presumed that the large range in $EF_{heating}$ estimations was mainly due to the larger variabilities in $F_{CO2}$ in lower
temperature, not only owing to the missing data in September and October especially when heating tendency started but also owing to relatively stratified condition, thus, larger vertical gradients in cold periods. Sensitivity test of $EF_{heating}$ was conducted by changing the threshold temperature, and 28.2% changes were drawn for $EF_{heating}$.

## 4 Conclusions

Direct $CO_2$ exchange quantified by EC method from urban center in Gwangju and estimated $EFs$ with land use characters
are summarized in Table 2. Even though $F_{CO2}$ measurements for September and October were missing, year round emissions of individual activities were able to be estimated from seasonal bias free $EFs$ extraction strategies; since the $EF_{traffic\_DOW}$ and $EF_{industry}$ were inferred from DOW and $EF_{heating}$ were extracted from the relation of $F_{CO2}$ with temperature, no specific monthly dependency was expected.






**Table 2.** Summary of emission factors (***EF***) and annual emissions for individual activities, only for the parameters inferred from day of week difference due to the limitations in data coverage for September and October.

| | Measured Flux | Traffic | Industry | Heat | Vegetation |
|---|---|---|---|---|---|
| ***EF***s (error) | 28.3 µmol m$^{-2}$ s$^{-1}$ (23%) | 0.017 µmol m$^{-2}$ s$^{-1}$ car$^{-1}$ (65%) | 103.3 µmol m$^{-2}$ s$^{-1}$ (42%) | 2.41 µmol m$^{-2}$ s$^{-1}$ °C$^{-1}$ (71%) | - (109%) |
| Area | 14.1 km$^2$ | 5.23 km$^2$ | 0.33 km$^2$ | 3.1 km$^2$ | 4.63 km$^2$ |
| Activity | - | 4023 car | 3,856 hr | 2,330 °C | - |
| Yearly emission | 554 kt CO$_2$ yr$^{-1}$ 39 kt CO$_2$ km$^{-2}$ yr$^{-1}$ 10.7 kg C m$^{-2}$ yr$^{-1}$ | 497 kt CO$_2$ yr$^{-1}$ 95 kt CO$_2$ km$^{-2}$ yr$^{-1}$ 25.9 kg C m$^{-2}$ yr$^{-1}$ | 21 kt CO$_2$ yr$^{-1}$ 63 kt CO$_2$ km$^{-2}$ yr$^{-1}$ 17.21 kg C m$^{-2}$ yr$^{-1}$ | 66 kt CO$_2$ yr$^{-1}$ 21 kt CO$_2$ km$^{-2}$ yr$^{-1}$ 5.83 kg C m$^{-2}$ yr$^{-1}$ | -29 kt CO$_2$ yr$^{-1}$ -6.28 kt CO$_2$ km$^{-2}$ yr$^{-1}$ -1.7 kg C m$^{-2}$ yr$^{-1}$ |

The year round total $CO_2$ emissions for individual activities were estimated from the information of hourly median of traffic counts, operations hours for normal days within the industrial area and areal coverage of heat related land use (details are in Table S1), and quantified as 497, 21, and 66 kt $CO_2$ yr$^{-1}$, respectively, thus the major source of $CO_2$ in a year in Gwangju is on-road vehicle and heat generation (85 and 11 %, correspondingly).

From these quantities with observed flux of $CO_2$, the plants influence on $CO_2$ exchange including photosynthesis and respiration can be estimated as the net balance of total emissions among all activities with observation; -29 (-27) kt $CO_2$ yr$^{-1}$

(only considering industry emission in EIA region), which is 5 % in total exchange in a year, was evaluated from $EF_{traffic\_DOW}$. Under the consideration of vegetation coverage in the fetch (4.63 km$^2$) as net $CO_2$ uptake area, the total carbon intake by vegetation was evaluated as 1.7 (1.6) kg C m$^{-2}$ yr$^{-1}$ which falls in the range of previous researches (0.28-2.45 kg C m$^{-2}$ yr$^{-1}$, (Febriani et al., 2018; Jo, 2002; Leu, 1990; Liu and Li, 2012; Rowntree and Nowak, 1991).

One should note that even though our evaluated $CO_2$ uptake strength agrees with literature, the uncertainty ranges are large

(109%), since it was quantified from the subtle balance among individual emissions with large variabilities. In addition, we did not account for the $CO_2$ emission from human respiration, thus we only provide lower limit of plant roles than actual.

Interestingly, with $EF_{traffic\_SLR}$, additional 432 kt $CO_2$ yr$^{-1}$ of $CO_2$ emission source(s) is(are) required to reconcile the observed $F_{CO2}$ which is unlikely the case for Gwangju. We conclude that this is a direct evidence in the importance of integrating all seasonal features in *EFs* estimation; missing data of specific month likely induced incorrect estimation due to

the failure in assimilating all seasonal features (fall characteristics in our case), thus seasonal bias free method i.e. $EF_{traffic\_DOW}$ is a more proper strategy in annual $CO_2$ emission estimation.





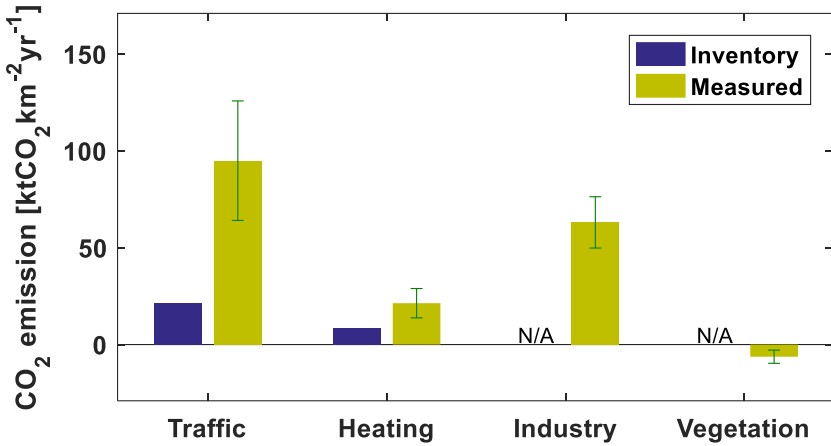

**Figure 8.** Comparison of $CO_2$ emissions between this study (dark yellow) and Gwangju emission inventory in 2017(dark blue). The whiskers represent uncertainties evaluated by Monte-Carlo bootstrap sensitivity test. Industrial emission and plant uptake are not available (N/A) from the inventory.

Based on our *EFs* estimations, both the annual $CO_2$ emissions of traffic and space heating (95 and 21 kt $CO_2$ km$^{-2}$ yr$^{-1}$) were more than 2.5 times higher than those of the emission inventory for Gwangju in 2017 (21.4 and 8.37 kt $CO_2$ km$^{-2}$ yr$^{-1}$, respectively; Figure 8). We presume that even with the systematic traffic counts with energy and fuel consumption monitoring, either the existence of fugitive $CO_2$ emissions or imperfection in *EF*s parameterization can be the reason for the underestimation. Thus, more efforts on the existing emission inventory validations are required. Unfortunately, we were not able to compare the annual $CO_2$ emission/uptake owing to the car manufacturing facility in EIA and urban vegetation, since the inventory was not compiled for it. Thus, site-specific emission inventories for individual facilities and validations are also needed in near future. In addition, more researches to understand the limited role of urban vegetation in $CO_2$ sequestrations are required as well as efforts to make this fact be known to the general public.

Even though we only provide lower limit estimations, our research results urge the necessity of emission inventory validation for establishing strategies that are more realistic to mitigate climate change. In this aspect, EC flux technique with careful analysis method is a useful tool even with relatively short-term period of measurements.

**Data availability**

All raw data can be provided by the corresponding author upon request.




**Authors contributions**

KEM took charge of the whole project starting from initiation to data analysis and wrote the manuscript with contributions from all co-authors. JM deployed the instrument and processed the data and BP and SL facilitated the instrument maintenance and edited the paper. KHK contributed on CFD modelling work.

**Competing interests**

The authors declare that they have no competing interests.

**Acknowledgements**

This work was supported by Gwangju Institute of Science and Technology Research Institute (GRI) of South Korea.

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
