# Peer review of "Insights on estimating urban CO2 emissions using eddy-covariance flux measurements"

_Atmospheric Chemistry and Physics, 2022_

## Author Comment (AC1)

Response to Reviewer 1

We thank the reviewers for all the helpful comments. This document includes the *complete text from the reviewer (**bold italic in black**)*, our responses to each comment, and the corresponding changes made in the revised manuscript with deleted parts crossed-out in red and added parts in blue. All of the line numbers refer to that of the original manuscript.

***Comments:***

***This study conducted the eddy covariance measurement at an urban area in Korea during a year for evaluating contributions from different emission sources, such as traffic, heating, industrial, and vegetation and validating an emission inventory. Since direct measurements of CO2 emissions are scarce in urban areas, the topic is interesting to potential readers and thus is suitable to the journal. However, due to several limitations in the presented work, I recommend substantial revisions.***

We thank the reviewer for all the helpful comments that led to the betterment of the paper. We tried our best to visit all the comments as follows.

Prior to the point-by-point response to all the comments, we would like to emphasize that there are no significant changes in our conclusions upon updating our analysis with the two key suggestions from reviewers 1 and 2, which are adding more stringent filtering criteria for turbulent flow statistics and using 90 % flux confinement in footprint analysis. All the related figures and numbers were updated accordingly in the main text and supporting information with tracked changes. Most of them that are related to the response to the comments are presented here but not all for the conciseness of this document, especially for the numbers and small changes.

***Main comments:***

***MC1. First, the authors did not state the objective of the study in the introduction. Consequently, I hardly justified the main conclusion, new finding, or hypothesis with the data. Furthermore, structure of the manuscript is not well organized. I strongly recommend that the results and discussion sections must be separated for deeper discussions.***

We reshaped the manuscript by updating the introduction to clearly emphasize the objective of this work as below.

(Line 25) As a mitigation strategy to climate change, most countries in the Knowing precise and accurate knowledge on anthropogenic carbon dioxide ($CO_2$) sources and their emission strengths($ES$s) is critical, since most countries in the world are living under the motto of reduction in greenhouse gas (GHG) emissions as a mitigation strategy to climate change. based on global consensus on seriousness and urgency in climate change. In order to realize this, many great deals of efforts are being invested not onlyon reducing $CO_2$ emissions from individual to international levels but also re-recognizing CO2 emissions as an economic concept via trading, taxing and capping, since this molecule comprises up to 65 % of radiative forcing among human-driven GHG emissions (NOAA, 2018). Thus precise and accurate knowledge on anthropogenic $CO_2$ sources and their emission strengths would be beneficial for establishing effective implementation strategies.

(Line 50) There are other indirect means to estimate $CO_2$ emissions, which can be compared with

existing bottom-up emission inventory, such as usage ~

(Line 57) Direct observation-based estimation methods are also used as tools for providing independent gauge for urban $CO_2$ emission in contrast to the existing bottom-up inventory.

(Line 90) However, even with the strength of direct monitoring of $CO_2$ vertical exchanges which has great potential to be linked with $ESs$ estimation,  less attempts have been made to infer $CO_2$ $ES$s from EC flux measurements compared to its spatiotemporal analysis. This limits the usage of EC flux measurements not only due to the heterogeneous nature of a city but also due to the requirement of long-term measurements for characterizing seasonally varying $CO_2$ emission features. Table 1  lists few studies which evaluated $ES$s for heating ($ES$s$_{heating}$) and traffic ($ES$s$_{traffic}$) from their $CO_2$ EC flux measurements with at least a year of coverage (except Ueyama and Ando, 2016; Liu et al., 2012; Vesala et al., 2008 and Velasco et al., 2005).  $ES$s$_{heating}$ were inferred from the linear relationship between observed $CO_2$ fluxes and temperatures over the heating degree days (HDD, Crawford and Christen, 2015; Kleingeld et al., 2018; Lietzke et al., 2015), while $ES$s$_{traffic}$ were mostly evaluated from simple linear relation (here after SLR; Järvi et al., 2012; Kleingeld et al., 2018; Lietzke et al., 2015; Matthews and Schume, 2022; Nemitz et al., 2002; Park and Schade, 2016; Velasco et al., 2009)  between traffic volumes and $CO_2$ fluxes.  Unless $CO_2$ emission characteristics within a year are sufficiently reflected thus fully cancelled out, this method is susceptible to the seasonal bias owing to variations in natural and anthropogenic $CO_2$ emissions such as changes in photosynthesis and respiration by urban vegetation, space heating, degree of incomplete combustion of vehicles, etc. In connection with this aspect, Helfter et al. (2011) described  varying $ESs_{traffic}$ among seasons with power correlation of traffic counts and $CO_2$ fuxes. Thus, more than a year of measurement is required to properly integrate the seasonal changes in $ESs$ with careful consideration in analysis for skewness in the number of observations for  each season. In some regions of the world, measured number of data imbalance among seasons are unavoidable due to the inherent natural variabilities especially vulnerable with open path sensors during rainy seasons.

(Line 104) To overcome this measurement period limitation in $ESs$ estimation using EC $CO_2$ flux measurement , we suggest an alternative analytical approach to extract $CO_2$ $E$$S$s among the differences in day of week (DOW) pairs from EC flux observations not only to minimize the seasonal bias but also to enable $E$$S$s evaluations with relatively short-term period than many years of measurements. This is presumed to be more suitable for urban monitoring where human-driven influences frequently change. From this practice, we were able to not only assess the $ESs$ of traffic, industry and heating but also  see the subtlety in inferring the magnitude of urban vegetation uptakes thus the importance of accurate estimation in $CO_2$ $ESs_{traffic}$ in addition to the comparison with existing emission inventory.

In addition, as suggested by the reviewer, the results and discussion sections are now separated into different sections. For the conciseness, we only include the section names here.

(Line 245) 3 Results

(Line 246) 3.1 Footprint and land use type analysis

(Line 265) 3.2 $CO_2$ measurement descriptions

(Line 314) 3.3 Emission  strengths ($ESs$) estimations

(Line 321) 3.3.1 $ES$_{traffic} emission

*MC2. As we know that urban flux measurements were not ideal especially for the instrumentation, I have concerns for the data quality of the eddy covariance measurements. The authors conducted the general quality controls according to Mauder and Foken (2004), but did not provide detailed information. In addition, authors used measured fluxes for all wind directions (Figs. 2, 4) although the flow distortion was expected (Fig. S1). Please show the flow statistics, such as sigma u, v, w per friction velocity for assuring the quality control.*

We would like to clarify that we only use the EC flux data which ranges from 45-225° due to the possible interruption in eddy character as described in the last part of section 2.1 (line 135 – 138).

Figure 2 (b) in the original version was the only part where we included the land use type in all directions to show the suitability of Gwangju city with diverse activities in the fetch, since Figure 2(c) and 2(d) only represent the subject fetches in further analysis. As shown in Figure 2(a), the wind sectors not used in this study, ranging from 225° to 45° in clockwise direction, are shaded in white.

However, we thank the reviewer for pointing out our mistake of not masking out the wind sectors we did not use in Figure 4 in the previous version. The figure has been updated without the data that fall in the sector which we did not analyze in this study.

[Figure]

**Figure 4.** Polar plots of (a) $CO_2$ mixing ratio and (b) $F_{CO2}$ centered at where the measurements were conducted.

As suggested by the reviewer, two more quality control schemes, $\sigma_w/u_*$ and $\sigma_u/u_*$, were adopted for additional filtering related to the near neutral condition. By following Stull (1988), 1.3 and 2.49 were used as the filtering criteria for $\sigma_w/u_*$ and $\sigma_u/u_*$, respectively. With these additional quality control criteria, 148 data points (marked as red and blue asterisks in the figure below) were removed.

[Figure]

Statements regarding this aspect were added as below. Figure S5 was also added showing the flow statistics of $\sigma_w/u_*$ and $\sigma_u/u_*$ with the final data used in this work.

(Line 157-162) Furthermore, thresholds of friction velocity  ($u_*$ >0.2) and flow statistics ($\sigma_w/u_*$ >1.3 and $\sigma_u/u_*$ >2.49, Figure S5) were  applied only for accounting sufficient turbulent condition, which results in 10.4 % data rejection. ~ With all the filtering processes, 38.6 % of data has been discarded.

[Figure]

Figure S5. Flow statistics of (a) $\sigma_w/u_*$ and (b) $\sigma_u/u_*$ with each filtering criteria (dotted black line) of 1.3 and 2.49 to only account for unstable condition in further analysis.

In addition, we added the spectral analysis result as shown in Figure S3 as a proof of legitimacy of using our filtered data set for further in-depth analysis. The main text was also updated accordingly.

[Figure]

**Figure S3. Co-spectral analysis results for the filtered data for in-depth analysis throughout the manuscript for (a) used wind sector (45°-225°) and (b) the other (225°-45°). Normalized co-spectra by individual flux of sensible heat (black circle), water (blue square) and $CO_2$ (red cross) against non-dimensional frequency (multiplied by measurement height, z, and normalized by wind speed, U) exhibit a slope of -4/3 as predicted in theory for (a) but not for (b).**

(Line 251) In addition, this is supported by the co-spectral analysis result of sensible heat, water and $CO_2$ fluxes shown in Figure S3 which exhibits a theoretical decaying rate of -4/3 in the inertial subrange only in 45°-225° wind sector

*MC3-1. Traffic CO2 emissions based on the two methods contained substantial uncertainties. First, the traffic counts were measured at a few points within (or outside?) the flux footprint; thus, the estimated emission factor (flux per car) should contain biases. The term emission factor should be inappropriate in this study.*

We mostly agree on the concerns about the possible biases owing to the difference in the traffic survey with the total number of vehicles in the fetch and the possible misleading due the usage of the emission factor terminology. Thus, we have replaced this terminology with emission strengths (*ES*s) and included the description that the traffic counts should be treated as a representative index of total traffic as in line 209. We hope that this concern would be spread out for all the works which relate the EC flux with traffic counts unless the traffic information is based on the accumulated total traffic in the fetch.

To test the legitimacy of using Gyesu as a representative index of the total number of cars within the fetch, we examined the variabilities of traffic counts among six survey sites in the fetch in various years starting from 2016 to 2019. As shown in Figure S8 with Table S1, no drastic variabilities among years with respect to Gyesu were observed. Especially for Sangmu Dist., the representative survey site for SGA, only 2 % difference with Gyesu was extracted for the year 2017 to 2018. To clearly address this point, the sentences below were added and updated in addition to Figure S8 and Table S1.

(Line 202) To address the possible bias of using traffic counts in one survey site, Gyesu, as a representative index of total number of cars within the fetch, we examined the variabilities of traffic counts among six survey sites within the fetch (locations are in Figure S6) starting from 2016 to 2019 (results are in Figure S8 with Table S1). Less than 10 % (20 %) changes in slopes with respect to Gyesu for the year of the study period (whole four years) were observed with good linearities (mostly, $R^2>0.8$, Table S1). Moreover, quite similar traffic counts were observed (<5 %) between Gyesu and Sangmu district, the representative survey sites for EIA and SGA, respectively, mainly indicating their similarities in traffic as urban center sites; especially for the year 2017 to 2018, their correlation coefficient was 0.98 (with $R^2$ of 0.81). Even though no large bias was expected, 2 % scaling down of Gyesu was used as the inferred instantaneous traffic information in SGA.

(Line 217) The traffic density  of Gyesu was used as a traffic index for EIA .

**Table S1. Traffic counts ratio of other survey sites with respect to Gyesu in the years 2016 to 2019. Slopes (intercepts) are extracted from linear regression.**

|  |  | 2016 | 2017 | 2018 | 2017-2018 | 2019 |
|---|---|---|---|---|---|---|
| Sangmu Dist. | slope(intercept) $R^2$ | 0.96 (-414), 0.74 | 0.99 (357) 0.88 | 1.04 (-179) 0.78 | 0.98 (270) 0.81 | 0.98 (136) 0.84 |
| Pungeum | slope(intercept) $R^2$ | 0.30 (-115) 0.80 | 0.28 (929) 0.76 | 0.30 (-116) 0.75 | 0.27 (328) 0.69 | 0.33 (-442) 0.88 |
| Gwangcheon | slope(intercept) $R^2$ | 0.54 (939) 0.87 | 0.48 (1768) 0.74 | 0.52 (1114) 0.81 | 0.52 (1035) 0.80 | 0.58 (634) 0.77 |
| Sangmu St. | slope(intercept) $R^2$ | 0.37 (88) 0.78 | 0.33 (554) 0.72 | 0.34 (151) 0.67 | 0.35 (150), 0.69 | 0.37 (-250) 0.78 |
| Yudoek | slope(intercept) $R^2$ | N/A | 0.70 (117) 0.96 | 0.77 (-1402) 0.98 | 0.74 (-1009), 0.97 | 0.72 (-476) 0.94 |

[Figure]

**Figure S8. Traffic counts variabilities among the survey sites near the city hall within the fetch in the years 2016 to 2019. The graphs of (a) to (e) show the diurnal patterns of other survey sites (triangle markers) starting from Sangmu Dist., Pungeum, Gwangcheon, Sangmu St. and Yudeok with respect to that of Gyesu (circle). The graphs of (f) to (j) represent the traffic counts relation of Gyesu with other stations, same sequence as (a) to (e).**

*MC3-2. Second, more seriously, human activity (e.g., commercial and business activities) could be closely related to traffic count at the diurnal and weekday/weekend scales. The simple regression or weekday/weekend statistics included not only traffic activity but also other human activity. This could overestimate the traffic CO2 emissions.*

We understand and admit that the original manuscript has not fully addressed the possible contribution of human activity in weekend(WE) and weekday(WD) differences. To clearly address it, we separately discussed the possibility of $CO_2$ emission by human respiration and commercial and business activities in section 4.2. Briefly, we are not expecting direct $CO_2$ emission difference by registered sectors of commercial and business areas but by human respiration due to the hourly difference in local population between WE and WD which is large enough to change the role of urban vegetation as a sink of $CO_2$.

(Line 341) ~ and thus an interesting topic for future investigations. Brief but more discussion on unaccounted $CO_2$ emission follows in section 4.2.

(Line 373) The possible high bias in $ES$s estimations can be attributed to the unaccounted $CO_2$ emission differences in weekend and weekday. The assumption of negligible $CO_2$ emission changes other than traffic and industry in weekly time frame (as mentioned in section 2.4.1 and 2.4.2) is an interesting topic for future investigations but discussed briefly in this section.

Commercial activities related $CO_2$ emission differences in DOW within our study area are hardly expected due to their opening hours, since the registered establishments within the fetch (Figure S15) are mostly neighborhood (36 %, i.e. restaurants, pubs, hair shops, theatre, etc.) and sales facilities (20 %, i.e. convenient stores, marts and department stores etc.). The transportation setups (13 %, express bus terminals and regular bus stops) also operate under the same schedule during weekend and weekday, and thus no changes in DOW is expected.

However, business related buildings (15 %) may differ in their $CO_2$ emission in DOW time frame, mainly related to heating and cooling tendency; cooling activities are based on electricity generated from outside of the footprint, while the heating, especially for the local heat generation facilities, may influence on $ES_{traffic\_DOW}$. However, as shown in Figure S14b, the slope variabilities among the $ES_{traffic\_DOW}$ with varying averaging months show no drastic changes (13 % ranges from 0.0141-0.0197) which indicate minor alteration in $ES_{traffic\_DOW}$ due to $CO_2$ emissions from heating in business-related buildings. Even though these small variabilities are influential enough to vary the role of vegetation in $CO_2$ from uptake to emission, no seasonal tendency was extracted in Figure S14b.

In addition, Figure S14, Figure S15 as well as section S9 in $SI$ have been added for the detailed descriptions on the registered building usage and floating demographic estimation as below.

S9. Registered building usage for commercial and business

The registered building usage for commercial and business area was investigated to examine the possible unaccounted $CO_2$ emission differences in weekday and weekend owing to these facilities. The data are available upon request through the Information Disclosure portal (https://www.open.go.kr/com/main/mainView.do) provided by the Korean government. The registered building usage with gross floor area from 2017 to 2018 was analyzed for commercial and business area within the fetch. The result is shown in Figure S15, where the major sectors were neighborhood (36 %), sales (20 %), business (15 %), and transportation facilities (13 %), respectively. Neighborhood facilities cover the class 1 and 2 types of living facilities that provide daily necessities and services necessary for general residential life such as daily necessities retailers, restaurants, coffee shops, beauty salons, hair shops, hospitals, bookstores, private academies, study cafés, pharmacies, pubs, theater, small markets, etc. Sales facilities refer to the larger markets, marts, convenience stores, department stores, etc. Transportation facilities are express bus terminals, regular bus stops, transit stations, etc.

[Figure]

**Figure S15. Status of registered buildings use for commercial and business activity in the study area.**

[Figure]

**Figure S14. Sensitivity test results of SLR (a, c) and DOW (b, d) for varying average period. Slopes (a, b) and intercept (c, d) were shown as bars with their fitting errors as whiskers.**

However, it is necessary to consider the difference in the number of floating populations during the morning rush hours on WE and WD. To assess the possible high bias in $ES_{traffic}$ due to human respiration difference, floating demographics data were considered. For that, the hourly difference in the number of people by DOW in the subject area based on communication mobile data was used. We added the following sentences in the main text in addition to section S10 in *SI* with Figure S16.

(Line 343) More importantly, we cannot rule out the possible high bias in $ES_{traffic\_DOW}$ owing to the $CO_2$ emission by human respiration from the changes in DOW floating population. To date, no readily available hourly population data exist, but it can be inferred from the monthly averages of hourly information of floating demographic (provided by Gwangju Big Data Integration Platform, https://bigdata.gwangju.go.kr) scaled with the data from private mobile communication carrier (provided by Magis based on SK Telecom mobile communication user data, https://www.magis.co.kr/) for weekday and weekend difference estimation (details are in *SI* S10 and Figure S16). From the inferred difference in the number of people during weekend and weekday (83,900 people within an hour in 7:00 to 10:00 time window) and $CO_2$ emission rate from human respiration (Prairie and Durate, 2007), 3.44 % of $F_{CO2}$ in DOW is attributed to human respiration itself which reduces the plant uptake rate to -0.73 kg C $m^{-2}$ $yr^{-1}$; this falls in the higher end of the ecosystem uptake rate (Baldocchi, 2014) and also induces -2.64 kt $CO_2$ $km^{-2}$ $yr^{-1}$ in yearly uptake rate. Thus, more works on direct $CO_2$ emission from human respiration should be investigated for more accurate estimation of vegetation's role.

In addition, the possible high bias in *ES*s estimations can also be attributed to the assumptions which cannot be proven by the available information. For example, we inferred the weekend traffic information based on Gyesu site survey result under the assumption that it represents the whole traffic situation in the fetch. The consistency among other survey sites were only tested for the weekdays rather than the weekend in Figure S7. Thus, real-time traffic data would be of practical help to not only validate our assumption but also reduce the possible errors in $ES_{traffic\_DOW}$.

S10. Registered building usage for commercial and business area

The number of variation in floating demography between weekend and weekday in hourly data is not open to the public, yet. To infer it, two different data sets were used. One is for the monthly average of total number of people within the fetch in hourly resolution while the other is for the day of week hourly ratio of number of people using a specific mobile phone carrier. The former was downloaded from the open data portal of Gwangju Big Data Integration Platform (https://bigdata.gwangju.go.kr/usr/main/goMain.rd) and the latter was acquired upon request to Magis (data of Chipyeong, where the city hall is located, https://www.magis.co.kr/), the data processing company which processes the mobile phone user data of SK Telecom Co, Ltd. Since the demographic data were available only after 2019, the data from May to July 2019 were used to match the data coverage of Magis. Due to the mismatch in periods, we limit this usage to examine the possible bias in $ES_{traffic}$ by human respiration rather than correction in $ES_{traffic}$.

Based on SK Telecom carrier, the average of weekend and weekday service population difference was 6,279 and the monthly average was 313,606 within the time window of 7:00-10:00. Thus, the total difference in weekend and weekday was estimated as 83,900 which can comprise up to 3.4 % in observed flux difference in weekend and weekday in that time window.

[Figure]

**Figure S16. Diurnal pattern in each day of week for the number of people who use the SK Telecom as their mobile service in Chipyeong dong where the city hall is located. The data collected were from May to July 2019.**

*MC4-1. For comparison to the inventory, many differences were shown in Fig. 8, but not were well discussed about the concrete reasons. Measured CO2 emissions were several times higher than those by the inventory. Such large discrepancy should be caused by fundamental problems with measurements and/or inventory. The authors must carefully discuss potential problems in the inventory with their calculation methods (e.g., how the inventory estimated emissions in detail).*

We would like to clarify that the current emission inventory has not been validated yet and must be refined to reflect reality. Thus, the importance of this manuscript is not only to raise the urgency for verifying the existing $CO_2$ emission inventory through various methods but also to emphasize the need for establishing real-time data sets of various parameters (i.e. traffic situations, floating population and registered land usage, etc.) as a firm digital infra for more accurate $CO_2$ emission estimation. We added more discussions on the potential cause of the difference between our estimation and inventory by adding descriptions on the methods on how the inventory has been constructed as below.

(Line 409) These large discrepancies are not surprising, as it has not been validated with any observations and is a subject to refinement in the near future. The inventory we compared to is an earlier version with many simplifications in it.

In terms of traffic related emissions, the current inventory is calculated based on the automobile fuel sale with emission factors of fuel type which is not necessarily identical to fuel consumption within the fetch, thus, it has been known for large uncertainties in estimations (International Climate & Environment Center, 2018; Kouridis et al., 2010). In addition, heating related $CO_2$ emission in the current inventory within the fetch is estimated by means of heat generation, such as district, individual and central heating. The $CO_2$ emission from the former one is estimated by integrated calorimeters and the latter two rely on the use of LNG (Liquefied Natural Gas) supplies via city gas pipelines. However, individual heating by LPG (Liquefied Petroleum Gas), briquette and biomass boilers are not accounted for. Thus, $CO_2$ emission from heating in the inventory is likely to be underestimated.

In addition, since $CO_2$ emissions are closely related to economic indicators, they are not aggregated by business location but only the national inventory for each company with reported figures from the individual makers are disclosed. In other words, unfortunately, it is not possible to directly compare the $CO_2$ emission of the automakers with our observation, since the reported $CO_2$ emission by industry in the Gwangju inventory does not include that of the factory. In addition, the national level inventory for those companies is being compiled by relying on self-reporting. However, in order to establish a national

level of $CO_2$ emission inventory with high-accuracy, it is necessary to first verify the figures and then aggregate them.

For urban vegetation, the inventory is compiled for parks, forests and street trees. Under the assumption of full coverage of park and impervious roads by vegetation, which is a lower limit in $CO_2$ uptake rate by vegetation since the park and roads are not fully covered with vegetation, -2.24 kt $CO_2$ km$^{-2}$ yr$^{-1}$ of $CO_2$ uptake happens in Gwangju (shown as empty box in Figure 8) which differs by 1~3 times from our estimation for with and without human respiration consideration (-2.64 ~ -6.28 kt $CO_2$ km$^{-2}$ yr$^{-1}$). For more accurate estimation in $CO_2$ sequester by urban vegetation, at least actual green cover aerial information should be disclosed.

*MC4-2. Furthermore, measured CO2 flux also contained missing values, but there was no description how gap-filling was conducted.*

We do not agree on the necessity for gap-filling since our measurement coverage is not sufficient enough to specify the seasonality of the site. Gap-filling without well-characterized seasonal changes will likely induce additional uncertainties. In the case of observation over a longer period of time that can fully characterize the seasonality in regional $CO_2$ emission, then a gap-filling method can be considered. The downside of this method especially for the urban center areas is that changes in $CO_2$ emission characteristics may occur even on long-term observations, which cannot be evaluated quickly.

This is where the other significance of this paper arises. As an alternative, data with instantaneously available information such as traffic counts difference in day of the week and temperature can be analyzed, thus, we propose a method that can analyze data without seasonal bias with relatively short-term observations.

*Specific comments:*

*SC1. Line 77: "thus easily covers a city scale". The statement is incorrect. Eddy covariance measurements even using a tall tower typically could not cover the entire city. Furthermore, given the heterogeneous nature of the city, spatial representativeness often hampered interpreting measured CO2 emissions.*

We rephrase the wording as below to address the limitations in scale coverage as well as in interpretations due to the heterogeneity of a city.

(Line 77) "~thus  covers a partial to full city scale"

(Line 91) "~limits the usage of EC flux measurements not only due to the heterogeneous nature of a city ~ "

*SC2. Table 1: Range of CO2 flux was vague in terms of their temporal coverage. Such information should be described with annual CO2 emissions or mean flux with specified period (e.g., daytime mean at the annual peak month).*

We updated the information on table1 with the annual $CO_2$ mean flux and the corresponding measurement period of each study.

***SC3. The unit of car seems to be strange, because the unit of traffic count should be car per period (e.g., car per sec, car per hour, or car per day).***

We intentionally dropped the unit of period to emphasize that the absolute amount of traffic volume from a survey is different from the total accumulative number of cars in the fetch as we have responded in 3-1. As described in line 208, the traffic volume from the survey has to be used as an intensity indicator of traffic volume and same frame should be used for scaling up estimation.

We admit that our discussion in the original manuscript was not enough to address this point. To clarify this, the sentences below were added on top of the already existing description in line 208.

(Line 208) One should note that the inferred traffic counts should be treated as an activity index which represents the relative amount of car fleet than the actual integrated number of cars within the fetch. Thus  simple comparison of $EFSs_{traffic}$ with other studies is not encouraged, since the degree of closeness in traffic volume with respect to the total traffic counts in a footprint likely varies among studies. To emphasize this aspect as well as the fact the traffic volume from the survey is an accumulation of the number of cars within the time frame rather than averages, we intentionally dropped the unit of period in traffic survey for the rest of the manuscript, but the same time frame should be used for yearly emission scaling up estimation.

In addition, caption in Figure 5 is updated as below.

**Figure 5.** Weekly diurnal patterns of (a) traffic volumes, (b) $CO_2$ mixing ratio and (c) $F_{CO2}$. Dotted marker with lines and area represent the mean and interquartile range of each factor. The time dimension in traffic volume has been dropped to indicate that the traffic count is not only an averaged value within in the time window but also an indicator of traffic intensity index which differs from the actual number of total cars within the fetch.

***SC4. Line 144: The equation of the covariance is too general and should be removed.***

Thank you for the suggestion and we somewhat agree on the usefulness of the equation throughout the manuscript We removed the covariance equation as equation 1 and the text was updated as below (line 182 and 183)

(Line 142) based on EC method, $F = \overline{w'c_x'}$  using the SmartFlux 2 (LI-COR) software, where ~

***SC5. Lines 175-189: As mentioned in the above major comment, DOW method contained uncertainties, because weekday/weekend differences in flux could be associated with differences in traffic as well as whether commercial sectors were open or not. Thus, DOW is influenced by CO2 emissions by commercial sectors.***

As addressed in MC3-2, since our fetch is located in the central area of the city, 56 % of the registered commercial establishments are neighborhood and sales facilities. Thus, there are no drastic differences expected. From the sensitivity test shown in Figure S14, no drastic changes in slopes of $ES_{traffic\_dow}$ were found with varying averaging month within the time window of 7:00-9:00 which indicate that there are no significant changes due to heating activity for business facilities.

However, as mentioned above, CO$_2$ emission by human respiration may differ due to the absolute difference in mobility in days in a week. Thus, we used the information from mobile phone data to estimate and reflect it in the data analysis as also described in MC3-2.

*SC6. Fig S3. Please change the color scale for easily distinguishing vegetation and non-vegetation areas. Currently, almost all sectors are colored green or yellow.*

The colors in Figure S3 in the original version is based on Korean administrative marking guidance and thus some of the residential areas showed up as greenish yellow since they contain vegetation cover.

However, as suggested by the reviewer, we changed the colors for better visualization and the same map is used in Figure 2(a) in the main text.

[Figure]

**Figure S6. Traffic counts monitoring sites of highway tolls (red balloon) and crossroads (yellow balloon) within the 3 km fetch from Gwangju city hall (magenta polygon) on the top of the satellite image and land use (left and right figure). One should note that the land use map shown here only represents the major land use type for the case of multi-purpose land in 3D. © NAVER**

[Figure]

**Figure 2.** (a) Footprint analyses on the land use map (red and blue lines by Kljun (2004) and Kormann and Meixner (2001), respectively, and purple line is their average) with fractions of land use types in percentage for (b) study area (45°-225°,

starting from the North as 0° in clockwise direction), (c) EIA (Eastern Industrial Area, 45°-100°), and (d) SGA (Southern Green Area, 100°-225°) sector. The white masked regions in (a) were not used. Black patched area in (a) represents the invisible region owing to the sensor height and location of the city hall. Land use types with fractions less than 1 % are not marked. © NAVER 2022

**SC7. Lines 217-218: Please explain more detailed information how the authors correct the traffic density.**

As described in 3-1, the traffic volumes of each representative survey site for EIA and SGA, Gyesu and Sangmu Dist. were compared and the difference especially for the years 2017 and 2018 were accounted for by inferring the traffic in SGA by 2 % scaling down those of Gyesu. As mentioned in MC3-1, sentences starting from line 202 and 217 were added as well as Figure S8 and Table S1 to give more information on this matter.

**SC8. Lines 226-233: I could not understand how HDD was used for the analysis, because temperature sensitivity was estimated with air temperature (Fig. 7). Add more information in details.**

We apologize for the unclear structure of the paragraph. HDD was used for yearly estimation. Thus, it should have been in section 2.4.3. To add more clarity, only the descriptions of $CO_2$ *ES* heating were retained in 2.4.2 and the HDD part was moved to section 2.4.3 as below.

(Line 213) To gauge the amount of $CO_2$ emission related to space heating during the low temperature season, the relation between temperature and $CO_2$ flux was used.  Both EIA and SGA data were considered for $ES_{heating}$ with time window of 10:00 to 14:00, when the $CO_2$ fluxes showed clear difference above and below 18 ℃ with light traffic condition. More specifically, the $ES_{heating}$ was estimated from the slope of $CO_2$ fluxes and temperatures under the temperature limit and the sensitivity tests were conducted by varying the threshold from 10 to 22 ℃ with 2 ℃ bins.

(Line 242) For the total $CO_2$ emission from heat,  $ES_{heating}$ was considered  with HDD of 2,330 ℃ in a year, where the HDD was estimated as similar to Kleingeld et al. (2018), by multiplying the heating temperature and the number of days when the temperature was below 18 ℃, assuming that the heating tendency stops once the 18 ℃ temperature is reached. For September and October in 2018, inferred HDD with second order regression was used (purple line in Figure S9).

**SC9. Lines 240-244: For upscaling CO2 emission to city scale, floor number of buildings must be considered for commercial and residential sectors in addition to aerial coverage.**

We do not agree with the reviewer's point on this, since the CO₂ *ESs* of heating are assessed from the projected areal coverage (the m² unit in measured $F_{CO2}$ refers to that) rather than the building surface area. For bottom-up estimation, the reviewer's point is true but the measured CO₂ flux should be treated as an effective emission of the unit area within the fetch. In the original version, it was incorrectly accounted for by only using the areal coverage of residential, commercial and other buildings in the fetch when we scaled it up to a year frame. Thus, proper corrections were made as below and the area information in Table2 was updated as well.

(Line 242) For the total CO₂ emission from heat, $EFs_{heating}$ was considered  with HDD of 2,330 ℃ in a year, where the HDD was estimated as by Kleingeld et al. (2018)  by multiplying the heating temperature and the number of days when the temperature was below 18 ℃, assuming that the heating tendency stops once the temperature reached 18 ℃. For September and October in 2018, inferred HDD with second order regression were used (purple line in Figure S9). Since the *ES*_heating_ was extracted as an effective CO₂ emission within the fetch, the total footprint area was considered (26.47 km², unlike *ES*_traffic_ and *ES*_industry_, *ES*_heating_ was directly inferred from the observed flux other than DOW differences, thus it should be treated as fluxes of the entire footprint area).

*SC10. Fig. 1b: Please show actual photos rather than deformed schematics because readers more easily understand the instrumentation based on the photo rather than the image.*

We added two photos of the instrument and its holding structures as Figure 1 (c) and (d) for side and top views; the figure caption was updated as well.

[Figure]

**Figure 1.** (a) Location of the study area in Gwangju, Korea, © Google Maps 2022, (b) instrumental set-up position on top of the Gwangju city hall building visualized in 3-dimensional view provided by the geospatial information open platform, Vworld, with real photos of side (c) and top (d) views.

*SC11. Lines 279-280. I am not sure how the authors wanted to explain using this statement. If the authors wanted to mention air storage or underestimates in turbulent fluxes, discuss more details in a quantitative manner.*

Our original intention was to describe the observed flux of $CO_2$ and explain the cause of higher values in cold season. To reflect this, we updated the sentence as below.

(Line 279-) As opposed to $F_{H2O}$, observed $F_{CO2}$ had higher mean and more variation in cold than warm season (34.56±21.55 vs 24.80±11.08, respectively), mainly due to the stronger  vertical concentration gradients reflecting the active changes in $CO_2$ emissions/uptake processes from various sources (i.e. space heating, respirations, incomplete combustions, etc.) during wintertime.

*SC12. Line 292-293. Polar plots in Fig. 4 are interesting, but were not described how is was conducted in the method section. Add detailed methods with relevant citations.*

We included descriptions on it as suggested but in the results section for the natural flow of our manuscript.

(Line 291) To investigate the source distribution, the polarPlot function which is available in R openair package (Carslaw and Ropkins, 2012) was used. Details on plotting mechanisms can be found elsewhere (Carslaw, 2019; Carslaw and Beevers, 2013; Grange et al., 2016; Uria-Tellaetxe and Carslaw, 2014), but briefly, means of The polar plots of $CO_2$ concentration mixing ratio and or flux were computed from the aggregated data set for wind speed and direction bins and smoothed for continuous surface fitting on polar coordinates using Generalized Additive Model (GAM). are As shown in Figure 4, where the color represents wind speed weighted $CO_2$ mixing ratio and $F_{CO2}$, wind speed increases from the center outward in radial direction thus the center of the plots indicates the condition of 0 m s$^{-1}$ wind speed. Enhanced $CO_2$ mixing ~

*SC13. Fig. 5 and lines 309-310. Weekday/weekend difference in cars seems to be marginal. Please explain how differences were statistically significant.*

We do not agree on that since the morning peaks in Saturday and Sunday were later and smaller compared to weekdays. To distinctively show the difference, we also provided Figure S12 in SI with descriptions in line 310-313. As a quantitative description, Wilcoxon rank sum test results were added on the text as below.
(Line 311) ~ two local maxima around 6:00-9 10:00 and 17:00-19:00 (Figure S12); the Wilcoxon rank sum test results show that morning (evening) traffics on weekend and weekday are significantly different with p-value of 0.0088 (0.0091).

Meanwhile, no significant difference in the traffic counts in 11:00-14:00 between weekday and weekend were observed as we intentionally set the time window. This was to minimize the error accumulation in $ESs_{industry}$ from $ESs_{traffic}$ as shown in Figure S12. As a quantitative description on this point, the sentence below was added.

(Line 313) Meanwhile, no statistically meaningful difference in weekday and weekend traffic counts were observed in the time window of 11:00 to 14:00 (p value: 0.1518) when we estimate for $ES_{industry}$.

**SC14. The unit of Fig. 5a should be [number per hour].**

As described in SC3, we deliberately dropped the unit for period to prevent the simple comparison of $EF_{traffic}$ with other studies. We would rather put the description of time unit in the caption rather than on the Y axis to reveal that those numbers are from hourly accumulation, since it is not only an averaged value of the time bin but also an indicator of traffic intensity index. Thus, the caption in Figure 5 has been updated as well as the text in line 218 as described in SC3.

**SC15. Furthermore, how are weekday/weekend differences in traffic count consistent among the traffic count sites? How did the range of inconsistency among the sites affect the estimates in the traffic CO2 emission?**

Unfortunately, sporadic traffic surveys were achieved with labor intensive investment. Thus, weekend traffic information was only available in one site, Gyesu crossroad, after 2019; surveys for other sites have been conducted only for the weekdays. We hope that this work can contribute to convince the readers of the value of collecting and sharing the real-time traffic data; some of the real-time traffic data available in the past have become more scarce, especially in city center due to maintenance issue. To clearly address this point, the following sentences were added in the text.

(Line 406) In addition, the possible high bias in $ES$s estimations can also be attributed to the assumptions which cannot be proven by the available information. For example, we inferred the weekend traffic information based on Gyesu site survey result under the assumption that it represents the whole traffic situation in the fetch. The consistency among other survey sites were only tested for the weekdays rather than the weekend in Figure S7. Thus, real-time traffic data would be of practical help to not only validate our assumption but also reduce the possible errors in $ES_{traffic\_DOW}$.

(Line 413) Thus, site-specific emission inventories for individual plant facilities and validations are also needed in the near future. In addition,  to understand the limited role of urban vegetation in $CO_2$ sequestrations, detailed level of knowledge such as better accuracy in $ES$s estimations even for the degree of human respiration alterations are required. For that, in situ traffic count information in addition to the hourly demographical information on floating population will be of practical help.

**SC16. Line 308: Be quantitative manners.**

Reflected as below.

(Line 308) weekday flux was higher than that of the weekend (weekday and weekend means are 26.34 and 15.59 µmol m$^{-2}$ s$^{-1}$, respectively).

**SC17. Line 364: I cannot understand how HDD was used.**

As we have responded in SC8, HDD was used for scaling up for yearly emission estimation and we rephrased the section as in SC8.

**SC18. Line 371: As mentioned above, further quantitative discussion is required.**

We revised our manuscript in a more quantitative manner as below. (The sensitivity test set up is in line 233 as "varying the threshold from 10 to 22 ℃ with 2 ℃ bins")

(Line 371) Sensitivity test of $ES_{heating}$ was conducted by changing the threshold temperature and 31.5% changes were drawn (median$\pm 1\sigma$ of 1.62$\pm$0.51 with interquartile of 0.73)

**SC19. Line 376: I cannot understand the rationale of this statement "were able to estimate from … strategies". Is this supported by the current data analysis?**

We intended to prove it from the descriptions of errors in slopes and variabilities in intercept between SLR and DOW in $ESs_{traffic}$ estimation as mentioned in line 327-330 and line 338-339; the errors in the slope and variabilities in intercept reduced in DOW than in SLR as the intercept variabilities and slope errors of fitted line in $ESs_{traffic}$ with SLR and DOW were $\pm 12.5$ μmol m$^{-2}$ s$^{-1}$, 92 % and $\pm 4.34$ μmol m$^{-2}$ s$^{-1}$, 65 %, respectively. Smaller variability and error are likely indications of removal in seasonal fluctuation.

As a direct proof of our statement, we performed sensitivity tests of slope changes with varying observation periods. A total of 5 groups (collection of 4-6 months each for similar number of available observations) were tested for SLR and DOW. As shown in Figure S14, the slope variabilities were larger in SLR (36 %) than in DOW (13 %). In addition, the slopes in SLR decreased in warmer than in colder season, which may be an indication of the influence of seasonality in $CO_2$ emission such as stronger $CO_2$ emission (uptake) in cold (warm) season.

[Figure]

**Figure S14. Sensitivity test results of SLR (a, c) and DOW (b, d) for varying average period. Slopes (a, b) and intercept (c, d) were shown as bars with their fitting errors as whiskers.**

To address the results of varying averaging month sensitivity tests, Figure S14 and the sentences below were added:

(Line 342) As a robustness test for seasonal influences for $ES_{traffic}$ in SLR and DOW method, slope variabilities with varying averaging months were tested (Figure S14). To ensure enough number of samples for averaging, collections of 4 to 6 months (for similar number of available observations for each group) were used. The slope variabilities were larger in SLR (36 %, for 1σ, ranges from 0.0011-0.0025) than that of DOW (13 %, ranges from 0.0141-0.0197), which is an additional evidence of reduced seasonality in DOW method. In addition, the slopes in SLR decreased as it progressed to warmer season which may be a further indication of influences of variations in $CO_2$ emission in season such as stronger $CO_2$ emission (uptake) in cold (warm) periods. From this analysis, we were able to show that DOW difference is a promising method for reducing seasonal influence in $F_{CO2}$, thus, $ES$s estimations with relatively short-term periods than many years of measurements are feasible.

In addition, we admit the fact that the parameters extracted from DOW may still have some seasonal features. Thus, we toned down our original expression as below,

(Line 376) ~ year round emission estimations of individual activities of $ES_{traffic\_DOW}$, $ES_{industry}$ and $ES_{heating}$ were possible able to be estimated from seasonal bias free $EFs$ extraction strategies; since the first two $EF_{traffic\_DOW}$ and $EF_{industry}$ were inferred from DOW differences which resulted in smaller seasonal bias and the latter one was $EF_{heating}$ were extracted from the relation of $F_{CO2}$ with temperature, thus no specific monthly dependency was expected.

***SC20. Line 392: The CO2 uptake by vegetation is too high. Please see Fig. 2 in Baldocchi (2014) which showed that range of annual CO2 uptake by natural or disturbed ecosystems.***

It is high compared to what Baldocchi (2014) showed in his Figure 2, which was construed mainly over vegetation canopy; however, there are other literature on urban EC flux of $CO_2$ measurements showing even higher uptake rate of $CO_2$ as mentioned in references in line 393.

As we re-examined the possible high bias in previous $ES_{traffic\_DOW}$ owing to human respiration, the vegetation uptake has lowered down to -0.73 kg C m$^{-2}$ yr$^{-1}$, and this falls in the higher end of the ecosystem uptake rate. However, we do not encourage any over-interpretation on this term since the estimated uncertainty is large (151%). To clarify this point, we added the sentences below in sections 4.1 and 4.2.

(Line 391) Under the consideration of vegetation coverage in the fetch (4.636.88 km$^2$) as net $CO_2$ uptake area, the total carbon intake by vegetation was evaluated as 1.7 (1.6)1.8 kg C m$^{-2}$ yr$^{-1}$ which falls in the range of previous researches from urban area (0.28-2.45 kg C m$^{-2}$ yr$^{-1}$, (Febriani et al., 2018; Jo, 2002; Leu, 1990; Liu and Li, 2012; Rowntree and Nowak, 1991), but higher than the annual $CO_2$ uptake by natural and/or disturbed ecosystem (0.16±0.28 kg C m$^{-2}$ yr$^{-1}$, Baldocchi, 2014). However, the large error in vegetation uptake (151 %) cannot be overlooked. We would like to clarify that the high bias of $ES$s for traffic and heating easily ended up to change the role of vegetation from $CO_2$ sink to source; roughly 6 and 13 % lower $ES_{traffic\_DOW}$ and $ES_{heating}$ induce $CO_2$ emission from vegetation. That is because the quantified vegetation uptake was from the subtle balance among individual emissions with large variabilities, which resulted in an uncertainty of 151 %, thus over-interpretation should not be made.

(second to the last paragraph in section 4.2) More importantly, we cannot rule out the possible high bias in $ES_{traffic\_DOW}$ owing to the $CO_2$ emission by human respiration from the changes in DOW floating population. There are no existing readily available hourly population data, but it can be inferred from the monthly averages of hourly information of floating demographic (provided by Gwangju Big Data Integration Platform, https://bigdata.gwangju.go.kr) scaled with the data from private mobile communication carrier (provided by Magis based on SK Telecom mobile communication user data,

https://www.magis.co.kr/) for weekday and weekend difference estimation (details are in *SI* S10 and Figure S16). From the inferred difference in the number of people during weekend and weekday (83,900 people within an hour in 7:00 to 10:00 time window) and $CO_2$ emission rate (Prairie and Durate, 2007), 3.44 % of $F_{CO2}$ in DOW is attributed to human respiration which reduces the plant uptake rate to -0.73 kg C m$^{-2}$ yr$^{-1}$; this falls in the higher end of the ecosystem uptake rate (Baldocchi, 2014) and also induces -2.64 kt $CO_2$ km$^{-2}$ yr$^{-1}$ in yearly uptake rate. Thus, more works on direct $CO_2$ emission from human respiration should be thoroughly investigated for future work.

**SC21. Line 420: Based on the current environment for open data science, "author upon request" seems to be insufficient. Please use public databases, such as KoFlux, FLUXNET, or other open databases.**

We wish we could be a part of those networks but our measurements were temporary. Unfortunately, the logistics situation is not positive to have this measurement again at the same site. However, due to the importance and urgency of this kind of work, our group is keeping up our effort to have prolonged urban $CO_2$ flux measurements in addition to the vertical and horizontal concentration measurements where the in-situ traffic information is available (i.e. Seoul in Korea). This is to not only reproduce this work but also move forward to estimate more accurate $CO_2$ emission in the near future.

**References**

Baldocchi, Measuring fluxes of trace gases and energy between ecosystems and the atmosphere – the state and future of the eddy covariance method, Global Change Biology, 20, 3600-3609, doi: 10.1111/gcb.12649, 2014

Carslaw, D. C. and Ropkins, K.: openair – An R package for air quality data analysis, Environ. Modell. Softw., 27–28, 52–61, https://doi.org/10.1016/j.envsoft.2011.09.008, 2012.

Carslaw, D.C. The openair manual — open-source tools for analysing air pollution data, Manual for version 2.6-6, University of York, 2019.

Carslaw, D. C. and Beevers, S. D.: Characterising and understanding emission sources using bivariate polar plots and k-means clustering, Environ. Modell. Softw., 40, 325–329, https://doi.org/10.1016/j.envsoft.2012.09.005, 2013.

Grange, S. K., Lewis, A. C., and Carslaw, D. C.: Source apportionment advances using polar plots of bivariate correlation and regression statistics, Atmos. Environ., 145, 128–134, https://doi.org/10.1016/j.atmosenv.2016.09.016, 2016.

Gwangju Big Data Integration Platform:https://bigdata.gwangju.go.kr/usr/mstr/goMstrChartUsr.rd., last access: 19 May 2022.International Climate & Environment Center: 2018 Greenhouse Gas Inventory Report in Gwangju Metropolitan City, Gwangju, Korea., 2018.

Kouridis, C., Gkatzoflias, D., Kioutsioukis, I., Ntziachristos, L., Pastorello, C., and Dilara, P.: Uncertainty estimates and guidance for road transport emission calculations, Tech. rep., Joint Research Center – Institute for Environment and Sustainability, 2010.

Uria-Tellaetxe, I. and Carslaw, D. C.: Conditional bivariate probability function for source identification, Environ. Modell. Softw., 59, 1–9, https://doi.org/10.1016/j.envsoft.2014.05.002, 2014.

Stull, R. B.: An Introduction to Boundary Layer Meteorology, Kluwer Academic Publishers, Dordrecht,

1988.

---

## Author Comment (AC2)

Response to Reviewer 2

We thank the reviewers for the comments and the updated manuscript is marked with deleted parts crossed-out in red and added parts in blue with **original text in black**. Greens are our responses and reviewer2's comments are *in black italic*.

Prior to the point-by-point response to all the comments, we would like to emphasize that there are no significant changes in our conclusions upon updating our analysis with the two key suggestions from reviewers 1 and 2, which are adding more stringent filtering criteria for turbulent flow statistics and using 90 % flux confinement in footprint analysis. All the related figures and numbers were updated accordingly in the main text and supporting information with tracked changes. Most of them that are related to the response to the comments are presented here but not all for the conciseness of this document, especially for the numbers and small changes.

*Comments:*

*"Insights on estimating urban CO2 emissions using eddy-covariance flux measurements"*

*Kyung-Eun Min et.al,2022*

*General comments*

*This manuscript try to quantify CO2 emission strengths of individual urban activities (vehicle, industry, heat generation et.al.) based on less than one year measurements with Eddy-Covariance (EC) method at Gwangju, Korea. The author estimated CO2 emission factors (EFs) of Traffic/Industry/Heat from the EC measurement, while the plants influence on CO2 exchange including photosynthesis and respiration can be estimated as the net balance of total emissions among all activities with observation (for the estimation of EF of vegetation). Based on their EFs estimations, they found that the annual CO2 emissions of traffic and space heating were more than 2.5 times higher than those of the emission inventory for Gwangju in 2017-2018.*

*1. However, this experiment setup and data are not reliable. The CO2 flux measuring system was installed on the helideck of the Gwangju city hall, so the building's effect on the EC measurement could not be ignoring. The results are not robust.*

We analyzed our data with caution due to the possible distortion caused by the city hall building itself. As described in the original manuscript, the CFD (computational fluid dynamics) model constrained for real architectural information of the building is used for careful data selection for further analysis.

As a proof of legitimacy in our measurement, we added the spectral analysis results for the wind sector used for *ESs* (emission strength) assessments which showed no drastic alterations in eddy break down with a theoretical slope of -4/3 as shown in Figure S3, which can be used as an evidence of reliability of our measurements for further in-depth analysis. The main manuscript has also been updated to refer Figure S3.

[Figure]

[Figure]

**Figure S3. Co-spectral analysis results for the filtered data for in-depth analysis throughout the manuscript for (a) used wind sector (45º-225 º) and (b) the other (225º -45º). Normalized co-spectra by individual flux of sensible heat (black circle), water (blue square) and CO2 (red cross) against non-dimensional frequency (multiplied by measurement height, z, and normalized by wind speed, U) exhibit a slope of -4/3 as predicted in theory for (a) but not for (b).**

(Line 251) In addition, this is supported by the co-spectral analysis result of sensible heat, water and $CO_2$ fluxes shown in Figure S3 which exhibits a theoretical decaying rate of -4/3 in the inertial subrange only in 45º-225 º wind sector.

*2. On the other side, there are lot of EC towers to measurement the Co2 flux in city since the beginning of 21st century, and some sites have collected more than 10 years dataset. CO2 emission factors (EFs) could not be used to other city as a universal parameter for the estimation of the annual co2 flux in the city.*

We are not sure about how we can find the "a lot of EC towers to measure the $CO_2$ flux in city since the beginning of 21st century, and some sites have collected more than 10 years of dataset" based on the first part of these comments. As far as we know, there are very limited number of existing publications on the urban $CO_2$ EC flux measurements as reviewer1 pointed out due to the difficulties not only in measurements but also in interpretation. In addition, the significance of our manuscript is not only limited to reporting the EC flux of $CO_2$ but also evaluating *ESs* (Emission Strengths) with site-specific characteristics. Even those existing long-term measurements, as mentioned on the last part of this comment, the *ESs* estimated from other cities cannot be applied to other cities as we also pointed out in our manuscript.

Additionally, we would like to emphasize the beauty of in-depth analysis with relatively short-term measurements in urban than the traditional long-term agricultural field application of EC technique, by suggesting a method to ease out the seasonal changes in $CO_2$ emission characteristics using the difference in day of the week pattern. Human activities in urban setting likely change over long-term period, like 10 years, especially in a developing country like Korea where active urban expansion happens. Its $CO_2$ emission pattern is altered through changes in various parameters such as traffic congestion condition related to changes in traffic volume, industrial activities in line with economic conditions, number of residents and their energy use, land use type (i.e. vegetation cover, bare land and open water, etc.), etc. Thus, in assessing $CO_2$ emission in urban setting, relatively short-term scale approach is more realistic and beneficial for establishing mitigation strategies in climate change.

On the other hand, we somewhat agree on reviewer2's point about "*CO2 emission factors (EFs) could not be used to other city as a universal parameter for the estimation of the annual*

*co2 flux in the city*". If that *ESs* refers to traffic, we cannot agree more as we also described in our manuscript. Thus, the word choice of *EFs* (Emission Factors) is not the best one as reviewer1 pointed out. Thus, we updated our terminology to *ESs*, hoping that our society will have more consensus on this issue. Meanwhile, for some of the *ESs* (i.e. heating, plant uptake), we may be able to use such as general parameters accompanied with detailed information (i.e. heating types and species by species vegetation cover).

(Line 103) In addition, long-term measurements of $CO_2$ fluxes are not necessarily linked with accurate *ESs* since urban utilization changes rapidly, in particular in developing countries.

*3. By the way, the model to simulate the co2 flux over city has been published (Jarvi, L., et al.(2019),JGR: Atmos.).*

We appreciate the suggested literature and we included it in our manuscript as below.

**(Line 56)** In addition, modelling attempts have also been made to study the variability and magnitude of carbon emissions and sinks in high spatial and temporal resolution (Jarvi et al., 2019). Specifically, the $CO_2$ model accounted for both anthropogenic and biogenic emissions over the city using combination of sources such as high-resolution airborne lidar-derived land use data and mobility data.

*4. This manuscript is not suitable to be accepted by ACP.*

We think we have fully addressed all the concerns of reviewer2 through our responses which are sufficient enough to highlight the significance of our work to contribute to the knowledge expansion in our atmospheric society through publication in ACP.

*Specific comments*

*1) L123-134 "The CO2 flux measuring system was installed on the helideck of the Gwangju city hall,……Our EC system was installed outside of inertial sublayer …and sufficiently lower than the planetary boundary layer.", is it correct?  The measurement is set in Gwangju city hall (90 m above the ground – building height: 85 m, helideck: 3 m and measuring system structure: 2 m), so it is not satisfied the guideline on the flux measurement in the city. The building's effect on the flow has large influence on the EC measurement.*

We thank the reviewer for pointing out our mistake. "~ outside ~" should be "~ inside ~". For better expression, we updated the sentence as below.

(Line 130) Our EC system was installed outside the roughness layer ~

*2) L166-167 "footprint boundaries were defined to confine 70% of average total flux during the measurement period.", usually we use the footprint boundaries to cover 90% of average total flux ".*

The degree of confined fraction is not a fixed number. The default value in SmartFlux 2 software is 70 %.

However, to reflect the concern on this aspect, we updated our analysis with footprint confinement of 90 % fluxes as shown in updated Figure 2. Following the analysis, most of the year round $CO_2$ estimations were updated in the main text in section 4 and Table 2 as well as in land use fractions in Figure 2 with corresponding descriptions.

(Line 166) Here, footprint boundaries were defined to confine 90% of average total flux during the measurement period. Detailed calculations and parameterizations are described in S3 of the *SI*/.

*3) L172-174 " To assess the quantitative contributions of the individual sources, the wind directions were split into two sectors; (1) the Eastern Industrial Area (EIA, 45º-100º) and (2) the Southern Green Area (SGA, 100º-225º), based on whether or not the fetch includes the automobile production plant and urban vegetation .", it is too simple to assess the quantitative contributions with two sectors division, due to the complex flow field in this city. So the EFs estimation has too large uncertainty.*

We are not sure what "complex flow field in this city" means. The topography of the city within fetches are pretty flat (please see more on the response related with your 4[th] comment). By considering the difference in weekend and weekday $CO_2$ fluxes for the same wind sector, the detailed variations are expected to be canceled out even if we are not sure about the existence of "complex flow" within our study area. In addition, as you can see from the polar plot in Figure 4, there are no distinct sources of $CO_2$ flux in EIA and SGA other than what has been expected.

*4) Gwangju city is located in a basin area, while there is a more than 1000m High Mountain in the east of the city. The local circulation due to the terrain should be occurred sometime during the season, and may be interaction with the urban heat island (UHI) in Gwangju city. So the co2 flux measurement may be also influence by the two circulation. This manuscript didn't consider any information on the topic.*

In fact, Gwangju is located in a semi-basin which is open in north-north-east to south-south-west. As shown in Figure S1, the Mudeungsan Mt. referred by reviewer2 is located more than 10 km away from the 90 % flux confined fetch. Within the footprint, the topography is relatively flat. This information has been added in the supporting information as S1.

(Line 11 in SI) S1. Topographical setting of study area

Gwangju is located in a semi-basin which is open in north-north-east and south-south-west direction. As shown in Figure S1, our study area is relatively flat (47±25 m, 3 hills up to 100 m in height are located close to the end of 90 % confined flux fetch where the radius of the footprint is close or less than 4 km); the small hills especially in southern area are lower than the mean building height throughout the fetch. The Mudeungsan Mt. and its foothills (1187 m ASL) are located more than 10 and 4 km away from the footprint.

[Figure]

**Figure S1. Topographical setting of study area with 90% flux confined footprints Kljun (2004), Kormann and Meixner (2001) and their averages in red, blue and bright green lines. The white V markers are the hills close to 100 m in height and the yellow balloon is the location of Mudeungsan Mt. Dotted dash pink line refers to the area shown in Figure 2(a) in the main text.**

In regard to the heat island, there are two possible alteration exits, one for the local circulation changes as the reviewer pointed out and the other for the storage. For the circulation alteration, as mentioned in our response in 3rd comments, taking the difference within a week time frame will lessen this concern. To add more clarity on this aspect, the sentence below was added.

(Line 181) In addition, the concerns about possible complex circulation owing to urban heat island phenomenon likely lessen in DOW difference.

In regard to heat accumulation near the surface, it likely matters to heat flux measurements than to $CO_2$. In order to quantify the heat flux exchange, storage corrections are necessary for urban setting. Heat island effects are not directly related to $CO_2$ exchange. However, we also mentioned the possible underestimation in $CO_2$ emission due to the limitation in vertical $CO_2$ concentration gradient measurements in line 21 and 319.

*By the way, the wind rose (daytime or nighttime) during 2017-2018 could not be found in the text.*

We added the wind roses in *SI* as Figure S10 with related descriptions as below.

(Line 273) Wind roses during the study period are shown in Figure S10 for wind distribution in the study area.

[Figure]

**Figure S10. Wind roses of filtered data from (a) all direction and (b) target wind sector for this study.**

**Reference**

Järvi, L., Havu, M., Ward, H. C., Bellucco, V., McFadden, J. P., Toivonen, T., Heikinheimo, V., Kolari, P., Riikonen, A., and Grimmond, C. S. B.: Spatial modeling of local-scale biogenic and anthropogenic carbon dioxide emissions in Helsinki, J. Geophys. Res.-Atmos., 124, 8363–8384, https://doi.org/10.1029/2018JD029576, 2019.